



**Ambient air quality in the Kathmandu Valley, Nepal during the pre-monsoon:**

**Concentrations and sources of particulate matter and trace gases**

Md. Robiul Islam[1], Thilina Jayarathne[1,a], Isobel J. Simpson[2], Benjamin Werden[3], John Maben[4], Ashley Gilbert[1], Puppala S. Praveen[5], Sagar Adhikari[5,6], Arnico K. Panday[5], Maheswar Rupakheti[7], Donald R. Blake[2], Robert J. Yokelson[8], Peter F. DeCarlo[3,9], William C. Keene[4], Elizabeth A. Stone[1,10]

[1]University of Iowa, Department of Chemistry, Iowa City, IA, USA
[2]University of California-Irvine, Department of Chemistry, Irvine, CA, USA
[3]Drexel University, Department of Civil, Architectural, and Environmental Engineering, Philadelphia, PA, USA
[4]University of Virginia, Department of Environmental Sciences, Charlottesville, VA, USA
[5]International Centre for Integrated Mountain Development (ICIMOD), Lalitpur, Nepal
[6]MinErgy Pvt. Ltd, Lalitpur, Nepal
[7]Institute for Advanced Sustainability Studies, Potsdam, Germany
[8]Department of Chemistry, University of Montana, Missoula, MT, USA
[9]Drexel University, Department of Chemistry, Philadelphia, PA, USA
[10]Department of Chemical and Biochemical Engineering, University of Iowa, Iowa City, IA, USA
[a]Now at: Purdue University, Department of Chemistry, West Lafayette, IN, USA

**Abstract:**

The Kathmandu Valley in Nepal is a bowl-shaped urban basin that experiences severe air pollution that poses health risks to its 3.5 million inhabitants. As part of the Nepal Ambient Monitoring and Source Testing Experiment (NAMaSTE), ambient air quality in the Kathmandu Valley was investigated from 11 to 24 April 2015, during the pre-monsoon season. Ambient concentrations of fine and coarse particulate matter ($PM_{2.5}$ and $PM_{10}$, respectively), online $PM_1$, inorganic trace gases ($NH_3$, $HNO_3$, $SO_2$, and $HCl$), and carbon-containing gases ($CO_2$, $CO$, $CH_4$, and 85 non-methane volatile organic compounds; NMVOC) were quantified at a semi-urban location near the center of the valley. Concentrations and ratios of NMVOC indicated that origins primarily from poorly-maintained vehicle emissions, biomass burning, and solvent/gasoline evaporation. During those two weeks, daily average $PM_{2.5}$ concentrations ranged from 30 to 207 µg m$^{-3}$, which exceeded the World Health Organization 24 hour guideline by factors of 1.2 to 8.3. On average, the non-water mass of $PM_{2.5}$ was composed of organic matter (48%), elemental carbon (13%), sulfate (16%), nitrate (4%), ammonium (9%), chloride (2%), calcium (1%), magnesium (0.05%), and potassium (1%). Large diurnal variability in



temperature and relative humidity drove corresponding variability in aerosol liquid water content, the gas-aerosol phase partitioning of $NH_3$, $HNO_3$, and $HCl$, and aerosol solution pH. The observed levels of gas-phase halogens suggest that multiphase halogen-radical chemistry involving both Cl and Br impacted regional air quality. To gain insight into the origins of organic

carbon (OC), molecular markers for primary and secondary sources were quantified. Levoglucosan ($1230 \pm 1153$ ng m$^{-3}$), 1,3,5-triphenylbenzene ($0.8 \pm 0.5$ ng m$^{-3}$), cholesterol ($3.0 \pm 6.7$ ng m$^{-3}$), stigmastanol ($1.4 \pm 6.7$ ng m$^{-3}$), and cis-pinonic acid ($4.5 \pm 0.6$ ng m$^{-3}$) indicate contributions from biomass burning, garbage burning, food cooking, cow-dung burning, and monoterpene secondary organic aerosol, respectively. Drawing on source profiles developed in

NAMaSTE, chemical mass balance (CMB) source apportionment modeling was used to estimate contributions to OC from major primary sources including garbage burning ($18 \pm 5\%$), biomass burning ($17 \pm 10\%$) inclusive of open burning and biomass-fueled cooking stoves, and internal-combustion (gasoline and diesel) engines ($18 \pm 9\%$). Model sensitivity tests with newly-developed source profiles indicated contributions from biomass burning within a factor of two of

previous estimates but relatively greater contributions from garbage burning (up to three times), indicating large potential impacts of garbage burning on regional air quality and the need for further evaluation of this source. Contributions of secondary organic carbon (SOC) to PM$_{2.5}$ OC included those originating from anthropogenic precursors for naphthalene ($10 \pm 4\%$) and methylnaphthalene ($0.3 \pm 0.1\%$) and biogenic precursors for monoterpenes ($0.13 \pm 0.07\%$) and

sesquiterpenes ($5 \pm 2\%$). An average of 25% of the PM$_{2.5}$ OC was unapportioned, indicating the presence of additional sources (e.g., evaporative and/or industrial emissions such as brick kilns, food cooking, and other types of SOC) or underestimation of the contributions from the identified source types. The source apportionment results indicate that anthropogenic combustion sources (including biomass burning, garbage burning, and fossil-fuel combustion) were the

greatest contributors to PM$_{2.5}$ and, as such, should be considered primary targets for controlling ambient PM pollution.

## 1. Introduction

According to the World Health Organization (WHO, 2016), 4.2 million (or 7.6% of all)
premature deaths globally during 2016 were linked to ambient air pollution. The majority of these premature deaths occurred in low- to middle-income countries in the South Asia, East Asia





and Western Pacific regions. The Kathmandu Valley in Nepal is home to more than 3.5 million residents who suffer from high levels of air pollutants, including particulate matter (PM), ozone ($O_3$), carbon monoxide (CO), and volatile organic compounds (VOCs) (Bhardwaj et al., 2018;

Kiros et al., 2016; Mahata et al., 2018; Putero et al., 2015; Sarkar et al., 2016; Wan et al., 2019) that are expected to have severe health impacts (Gurung and Bell, 2013).

Effective mitigation of air pollution requires understanding the major contributing sources. PM emissions contain molecular and elemental fingerprints that reflect the material from which the PM was generated and the process(es) by which it formed. For organic aerosol sources, these

chemical fingerprints include molecular markers that are defined as chemical species unique to a PM source category (Schauer et al., 1996). Well-established molecular markers for primary (direct emissions) and secondary (produced in the atmosphere from reactive precursors) sources are summarized in Table 1. These species can be used to identify sources of PM in ambient air both directly and through source apportionment modeling.

The Nepal Ambient Monitoring and Source Testing Experiment (NAMaSTE) was initiated in 2015 to characterize widespread and under-sampled combustion sources in Nepal. Source characterization measurements included trace gases (Stockwell et al., 2016) and particulate matter (Goetz et al., 2018; Jayarathne et al., 2018), as well as optical properties of aerosols (Stockwell et al., 2016). The characterized sources included brick kilns, garbage

burning, power generators, diesel groundwater pumps, idling motorcycles, cooking stoves, crop residue burning, and open burning of biofuels. As part of NAMaSTE, a regional monitoring station was also installed to probe the relative contribution of these sources to ambient air quality. In addition, new emissions data is being incorporated in to regional air quality models for the region (Zhong et al., 2018).

High daily average concentrations of $PM_{2.5}$ (up to 160 µg m$^{-3}$) (Shakya et al., 2017a) and $PM_{10}$ (up to 579 µg m$^{-3}$) have been documented in the Kathmandu Valley (Giri et al., 2006). Satellite-derived aerosol optical depth study indicate substantial increases of particulate loading in the Kathmandu Valley and nearby background sites over the past 15 years (Mahapatra et al., 2019). Major components of regional PM include black carbon (BC; 17%), sulfate (17%), and

ammonium (11%) measured in $PM_{2.5}$ (Shakya et al., 2017a), and organic carbon (OC; 23%) and nitrate (2.5%) measured in $PM_{10}$ (Kim et al., 2015). Carbon isotope analysis of bulk aerosol sampled during winter of 2007-2008 in the Kathmandu Valley suggested that a major fraction of





particulate OC originated from primary sources (69%), particularly local fossil fuel emissions (39%) (Shakya et al., 2010). A recent carbon isotope study observed that fossil fuel contributed

67% of the black carbon during April 2013 in the Kathmandu Valley (Li et al., 2016).

A earlier chemical mass balance (CMB) source apportionment study at Godavari, at the southeast edge of the Kathmandu Valley identified sources of $PM_{2.5}$ OC as biomass burning (21%), fossil fuel combustion (7%), and secondary organic aerosol from biogenic precursors (SOA; 3%) (Stone et al., 2010). However, the relative contributions of biomass and fossil fuel to

EC were highly uncertain due to large variability in EC emissions with respect to combustion efficiency and air-to-fuel ratios. A significant fraction of $PM_{2.5}$ OC (54-84%) in that study was unapportioned suggesting significant contributions from other primary and secondary sources. Assessments of PM sources in this region were challenging due in part to poorly-characterized emissions from brick kilns, garbage burning, and local industries (Stone et al., 2010).

Anthropogenic SOA was also identified as a likely source of $PM_{2.5}$ that has not previously been apportioned (Stone et al., 2012).

The primary goal of the study reported herein is to characterize the composition of ambient gases and PM in the Kathmandu Valley, Nepal and apportion major sources based on new knowledge of source-specific emissions within the region. Our specific objectives are to: 1)

quantify atmospheric loadings of volatile carbon-containing compounds ($CO_2$, CO, $CH_4$, and 85 non-methane volatile organic compounds; NMVOC), inorganic trace gases ($NH_3$, $SO_2$, $HNO_3$, HCl, total volatile inorganic Br), and PM mass in the Kathmandu Valley during pre-monsoon season; 2) chemically characterize the major carbonaceous and ionic constituents of PM; and 3) apportion organic carbon (OC) to its sources using CMB modeling with region-specific source

profiles when available. This work is designed to contribute to advancing the understanding of the role of combustion and other major pollution sources in South Asia and their effects on air quality.

## 2.  Methods

### 2.1. Site description

Ambient air was sampled at Bode (27.689$^o$ N, 85.395$^o$ E, 1345 m a.s.l), which is a semi-urban location close to the geographic center of the Kathmandu Valley. Bode was the measurement supersite during the Sustainable Atmosphere for the Kathmandu Valley-Atmospheric Brown



Clouds (SusKat-ABC) international air pollution measurement campaign (Mahata et al., 2017;
Mahata et al., 2018; Sarkar et al., 2016). Bode is located in Madhyapur-Thimi Municipality and
three major cities are located nearby: Kathmandu Metropolitan City to the west, Lalitpur
Metropolitan City to the southwest, and Bhaktapur Municipality to the southeast. The Bode
supersite was located in a newly developing suburban area that started with a grid of streets
placed across what were agricultural fields, with a gradual filling in of houses on individual
plots, while a lot of fields and empty plots still remain. Nearby the Bode site are agricultural
fields, Bhaktapur Industrial Estate with several small pharmaceutical, plastic, and metal
industries, and about 19 brick kilns located within 5 km to  the east and south-east of the site.
Meteorological conditions (temperature, relative humidity (RH), barometric pressure, global
radiation and precipitation) were measured at Bode with data averaged every five minutes. From
23 to 26 April, on-site meteorological measurements were not available so meteorological
conditions recorded at the Tribhuvan International Airport in Kathmandu, ca. 4 km to the west of
Bode, were used instead.

### 2.2. PM and reactive trace gas sample collection

A medium volume sampler (URG-3000 ABC) was placed on the rooftop of a five story
building at Bode, approximately 15 m (50 ft.) above the ground. PM samples were collected
from 11 to 24 April 2015, during daytime (8:00 AM to 5:30 PM) and nighttime (6:00 PM to 7:30
AM) intervals. $PM_{2.5}$ was sampled downstream of two 2.5 µm sharp-cut cyclones and $PM_{10}$ was
sampled downstream of a 10 µm sharp-cut impaction plate. Both air streams were split to collect
four discrete PM samples in each size bin (total of 8 samples per time interval), at nominal flow
rates of 7.4 L $min^{-1}$ each. The flow rate through each channel was measured before and after the
sample collection with a calibrated rotameter (Gilmont Inst., Barrington, IL). PM in each size
range was sampled on three 47 mm quartz fiber filters (QFF; Tissuquartz, Pall Life Sciences,
East Hills, New York) and one 47 mm Teflon filter (Teflo Membrane, 2.0 µm pore size, Pall Life
Sciences). QFF were pre-cleaned by baking at 550 $^{o}$C for 5 hours and 30 min to remove organic
species (Stone et al., 2007).  Following collection, exposed PM samples were transferred to
polystyrene petri dishes lined with pre-cleaned aluminum foil, capped, sealed with Teflon tape,
stored frozen in sealed polyethylene bags, and shipped to the University of Iowa for analysis.





Soluble reactive trace gases were sampled downstream of two of the PM$_{2.5}$ QFF filters during
daytime and nighttime periods using a filter pack technique (Bardwell et al., 1990; Keene et al.,
2009; Pszenny et al., 2004). Total volatile NO$_3$ and Cl (dominated by and hereafter referred to as
HNO$_3$ and HCl, respectively), SO$_2$, and total volatile inorganic Br (Br$_t$) were sampled using a
three-stage, 47-mm, Teflon filter pack housing configured with a QFF for PM collection (as
described above) followed by tandem rayon filters (Schleicher and Schuell, 8S) impregnated
with a solution of 10% potassium carbonate (K$_2$CO$_3$) and 10% glycerol. NH$_3$ was sampled in
parallel using an otherwise identical filter pack configured with tandem rayon filters impregnated
with a solution of 10% citric acid (C$_6$H$_8$O$_7$) and 10% glycerol.

In total, 27 sets of ambient PM and reactive gas samples were collected. Field blanks were
prepared every fifth sampling period by loading, mounting, recovering, unloading, and
processing filter housings using otherwise identical procedures as those for samples but without
pulling air through them. All filter housings and samples were loaded and unloaded using clean-
handling procedures. Impregnated filters were stored in polystyrene petri dishes that were
capped, sealed with Teflon tape, stored frozen in polyethylene bags, and shipped to the
University of Virginia for analysis.

Submicron PM (PM$_1$) was characterized in parallel for non-refractory constituents using a
High-Resolution Aerosol Mass Spectrometer (HR-AMS) (DeCarlo et al., 2006). The inlet for
the AMS was located within 5 m of the URG sampler. Samples were size selected through a
PM$_{2.5}$ cyclone, and dried to below 30% RH by a counter flow nafion dryer. The AMS measures
mass and composition of non-refractory PM$_1$ at 1-minute time resolution. Calibrations were
undertaken for alignment, mass (ionization efficiency) and particle sizing. Frequent, intermittent
power outages at Bode interrupted AMS operations  necessitating long restart times and
significant losses of sampling. Due to associated data losses only 10 of the filter samples aligned
with concurrent AMS data across the entire periods: 16 April nighttime to 17 daytime, 18
daytime to 21 daytime, and 22 daytime. An in-depth analysis of the HR-AMS data will be the
subject of a forthcoming paper, but general observations will be used here to provide higher time
resolution context for the filter measurements discussed in detail.

**2.3. Whole Air Samples**





Whole air samples (WAS) were collected from 16 to 24 April 2015 and analyzed for $CO_2$, CO
$CH_4$ and 85 non-methane volatile organic compounds (NMVOCs) using multi-column gas
chromatography (Simpson et al., 2011; Stockwell et al., 2016). The WAS analytical details,
including calibration procedures, are described in detail in Simpson et al. (2011). While the
WAS sampling was cut short by the Gorkha earthquake on 25 April 2015, and only nine samples
were collected, the limited WAS sampling still provides useful context for VOC levels and
sources in the area.

### 2.4. PM$_{2.5}$ mass measurement

PM mass was measured on Teflon filters as the difference between post- and pre-sampling filter
masses. Prior to mass measurements, filters were conditioned for 48 hours in a temperature ($22 \pm$
$0.5\ ^oC$) and humidity ($34 \pm 12\%$) controlled environment. Masses were measured in triplicate
using an analytical microbalance (Mettler Toledo XP26). PM mass per filter was converted to
mass concentration ($\mu g\ m^{-3}$) using sampled air volume after field blank subtraction. The
analytical uncertainties in the mass measurements were calculated following Jayarathne et al.
(2018). The PM$_{2.5}$ data for the nighttime periods of April 12 and April 13 were excluded due to
sampling errors and filter damage, respectively.

### 2.5. Organic and elemental carbon measurement

Organic carbon (OC) and elemental carbon (EC) were measured following the National Institute
for Occupational Safety and Health (NIOSH) 5040 method (NIOSH, 2003) using 1.0 cm$^2$ filter
punches from sampled QFF (Sunset OC-EC Aerosol Analyzer, Sunset Laboratories, Tigard, OR)
(Birch and Cary, 1996). OC data were field blank subtracted, while EC was not detected on field
blanks. The uncertainties in OC and EC measurements were calculated following Jayarathne et
al. (2018).

### 2.6. Analysis of particulate inorganic ions
Water-soluble inorganic ions in PM were extracted in 5 mL deionized water and analyzed by ion
exchange chromatography (IC) with conductivity detection (Dionex-ICS 5000), with details of
the analytical method, uncertainties, and detection limit calculations provided by Jayarathne et
al. (2014). Unusually high concentrations of $Na^+$, $Mg^{2+}$, $Ca^{2+}$, and $F^-$ were observed in the field





blanks collected on 15 April and PM samples collected from 15 to 17 April, indicating likely
       contamination of these samples. Thus, concentrations of these ions during this time period were
       not reported and were excluded in the calculation of average concentrations.

### 2.7. Analysis of reactive trace gases

Exposed rayon filters were extracted under sonication in 5 mL deionized water and analyzed by
       IC (Dionex-ICS 3000, dual channel).  The anion channel was configured with Dionex guard
       (AG-4A 4x50mm) and analytical (AS-4A 4x250mm) columns and a Dionex electolytically
       regenerated suppressor (AMMS).  The cation channel was configured with Dionex guard
       (IonPac CG16: 5 x 50mm) and analytical (IonPac CG16: 5 x 250mm) columns and a Thermo
Scientific Dionex electrolytically regenerated suppressor (CERS 500: 4mm). Standard solutions
       were matrix matched with sample extracts. Analytical results for samples were blank corrected
       based on median concentrations of analytes measured in extracts of field blanks.  Independent
       analyses of tandem rayon filters indicate that all analytes were sampled by the upstream filters at
       average efficiencies of greater than 98%.  Average detection limits estimated following Keene et
al. (1989) were 0.66 ppbv for $NH_3$, 0.065 ppbv for $HNO_3$, 0.035 ppbv for $HCl$, 0.18 ppbv for
       $SO_2$, and 0.014 ppbv for $Br_t$.   Estimated precisions based on replicate analyses were
       approximately ±5% of measured mixing ratios or ±0.5 times estimated detection limits (DL),
       whichever were the greater absolute values. Due to suspected contamination of filter samples
       collected during 15 to 17 April 2015 (described in section 2.6), results for these samples were
excluded from the reported data set.

### 2.8. Thermodynamic calculations

       Aerosol liquid water contents (LWC), activity coefficients, and the partitioning between ionized
       and solid aerosol constituents were calculated using E-AIM Model IV, which considers particles
comprised of $NH_4^+$, $Na^+$, $SO_4^{2-}$, $NO_3^-$, $Cl^-$, and $H_2O$   (Friese and Ebel, 2010)
       (http://www.aim.env.uea.ac.uk/aim/aim.php). E-AIM requires that the input data for ionic
       composition be balanced on an equivalent basis (i.e., Σ cations = Σ anions). Unmeasured ionic
       constituents (e.g., carboxylic anions such as oxalate), ionic constituents that were measured but
       are not considered by E-AIM (e.g., $K^+$, $Mg^{2+}$, and $Ca^{2+}$), and random analytical errors introduce
minor ion imbalances in the subsets of input data. To balance an anion deficit for a given sample,





the input concentrations of $SO_4^{2-}$, $NO_3^-$, and $Cl^-$ were increased in proportion to their measured concentrations on an equivalent basis. Similarly, to balance a cation deficit, concentrations of $NH_4^+$ and $Na^+$ were increased in proportion to their measured concentrations. Because the ionic compositions of aerosol sampled were dominated by $NH_4^+$, $SO_4^{2-}$, $NO_3^-$, and $Cl^-$, these

adjustments in measured concentrations were relatively minor (typically <15% for a given analyte). Sensitivity studies indicate that alternative approaches to charge balance input data for E-AIM yield similar results (e.g., Young et al. (2013)).

For each sample, the input data included the measured (or adjusted as described above)

concentrations $NH_4^+$, $Na^+$, $SO_4^{2-}$, $NO_3^-$, and $Cl^-$ and the corresponding temperature and RH averaged over the sampling interval. Model output used for subsequent calculations included aerosol LWC; activity coefficients for $NH_4^+$, $NO_3^-$, and $Cl^-$; and, for mixed-phase particles, the partitioning of $NH_4^+$, $NO_3^-$, and $Cl^-$ between dissolved and solid phases. E-AIM simulated three distinct regimes: 1) at RHs greater than about 75%, the aerosol was completely deliquesced (i.e.,

virtually all $NH_4^+$, $NO_3^-$, and $Cl^-$ were ionized), 2) at RHs less than about 60%, particles existed entirely as solids with only tightly bound water molecules and negligible LWC, 3) at RHs between about 60-75%, constituents partitioned between dissolved and solid (primarily $(NH_4)_2SO_4$) phases. Extraction of aerosol samples into dilute aqueous solutions prior to analysis would have dissolved any solid phases that were originally present in particles at ambient LWCs.

Consequently, the measured concentrations of ions in dilute aerosol extracts correspond to the total concentrations (ionized + solid) that existed in ambient aerosol prior to extraction. In these cases, the ratios of ionized to total (ionized + solid) $NH_4^+$, $NO_3^-$, and $Cl^-$ predicted by the model were used to calculate the fractions of the measured concentrations that were ionized in aerosol solutions at ambient RHs.


Equilibrium hydrogen ion activities for PM$_{2.5}$ and PM$_{10}$ during each sampling interval were calculated based on the measured phase partitioning and associated thermodynamic properties of compounds with pH-dependent solubilities (HNO$_3$, NH$_3$, and HCl) following the approach of Keene and Savoie (1998). Briefly, using HNO$_3$ as an example, the equilibrium

$$HNO_{3\,(g)} \overset{K_H}{\leftrightarrow} HNO_{3\,(aq)} \overset{K_a}{\leftrightarrow} H^+ + NO_3^-$$



was evaluated on the basis of simultaneous measurements of gas-phase $HNO_3$ mixing ratios and particulate $NO_3^-$ concentrations in air; temperature-adjusted Henry's Law ($K_H$) and acidity ($K_a$) constants for $HNO_3$ (Young et al., 2013); and aerosol LWCs, $NO_3^-$ activity coefficients, and the fractions of measured particulate $NO_3^-$ concentrations that were ionized as predicted by E-AIM (described above).


Most mixing ratios (75%) for volatile inorganic Cl were less than the estimated detection limit and the balance of measurements were near the detection limit, which constrained data interpretation based on the phase partitioning of HCl. However, both $NH_3$ and $HNO_3$ were present at concentrations well above the corresponding detection limits and, as described in more

detail below, the measured phase partitioning of these gases yielded paired estimates of aerosol solution pHs that agreed well (generally within ±0.1 to ±0.3 pH units). Using the mean $H^+$ activity for each set of these paired estimates coupled with the corresponding $Cl^-$ concentration for $PM_{2.5}$, aerosol LWC predicted by E-AIM, meteorological conditions, and the thermodynamic properties of HCl, the equilibrium mixing ratio for HCl during each sampling interval was

calculated directly (hereafter referred to as $HCl_{calc}$).

Results based on the above approach are subject to several inherent limitations. First, $PM_{2.5}$ and $PM_{10}$ were collected in bulk over relatively long (nominally 12-hour) sampling intervals, which could have driven artifact phase changes of compounds with pH-dependent solubilities and

associated bias in the measured gas and particle phases species concentrations. For example, based on their thermodynamic properties, $NH_3$ partitions preferentially with the more highly acidic, typically smaller longer-lived aerosol size fractions whereas $HNO_3$ and HCl partition preferentially with less acidic, typically larger shorter-lived aerosol size fractions (Keene et al., 2004; Young et al., 2013). When chemically distinct particles are mixed together in a bulk $PM_{2.5}$

or $PM_{10}$ sample, the pH of the bulk mixture typically differs from that of the aerosol size fractions with which these gases partitioned preferentially in ambient air, which drives artifact volatilization of $NH_3$ as well as $HNO_3$ and HCl (Keene et al., 1990; Young et al., 2013). Similarly, mixing chemically distinct particles sampled at different times over sampling intervals could drive artifact volatilization or condensation of gases. Exposing time-integrated aerosol

samples to gas phase mixing ratios that vary over sampling intervals can also drive artifact phase



changes. Because of their large surface-to-volume ratios, sub-µm-diameter particles rapidly equilibrate (in seconds to minutes) with interstitial gases and, consequently, are typically at or near thermodynamic equilibrium with the gas phase (Meng and Seinfeld, 1996). In contrast, larger particles equilibrate more slowly and may exhibit finite phase disequilibria (e.g., Keene

and Savoie (1998)). The assumption of thermodynamic equilibrium on which this analysis is based may not be entirely valid for constituents associated primarily with larger aerosol size fractions. In addition, the use of average values to characterize meteorological conditions over sampling intervals does not capture the full range of variability of the multiphase system.  On most days, RHs fell to minima less than 60% during daytime and increased to maxima greater

than 75% at night (Fig. 3a). Consequently, based on E-AIM, the actual hydration state of particles varied from virtually dehydrated to virtually completely deliquesced conditions over most diel cycles.  Presumably, between collection and recovery, the compositions of aerosol deposits on sample filters exposed to ambient air also evolved in response to changing RH and temperature. If so, meteorological conditions at recovery times rather than those averaged over

sampling intervals may be more appropriate metrics for evaluating phase partitioning and pH. Finally, the thermodynamic properties of gases considered herein (particularly HCl) are associated with non-trivial uncertainty that contributes to variability in results as discussed by Young et al. (2013).  Despite these limitations, results provide useful insight regarding major processes that modulate gas-aerosol phase partitioning of major atmospheric constituents,

associated aerosol acidities, and pH-dependent chemical transformations in the Kathmandu Valley.

### 2.9. Extraction and analysis of organic species in PM$_{2.5}$ by gas chromatography mass spectrometry

All glassware used in solvent extraction was pre-washed with ultra-pure water and baked at 500

°C for 5.5 hours. Based on the OC loading on the filters, one or more QFF for each time period was extracted following the procedure described in Al-Naiema et al. (2015). Organic species were analyzed using gas chromatography coupled to mass spectrometry (GC-MS, Agilent Technologies GC-MS 7890A) equipped with an Agilent DB-5 column (30 m x 0.25 mm x 0.25 µm) and electron ionization (EI) source with a temperature program described in Stone et al.

(2012). All the measured species were field blank subtracted and analytical uncertainties of the



measurements were propagated from the standard deviation of the field blanks and 20% of the measured concentration to conservatively account for compound recovery from QFF.

## 2.10. Quality control in chromatographic measurements of PM

For every five ambient samples, one lab blank, one field blank, and one spike recovery sample were analyzed for both organic species and inorganic ions analysis. Spike samples were prepared from blank filters spiked with known concentrations of analytes. These quality control samples were extracted simultaneously with ambient samples. Spike recoveries, reported as percent, were calculated as the quotient of the lab blank-corrected measured concentration and spiked

concentration. Spike recoveries of all the reported chemical species were within ±20% for the organic species and ±10% for the inorganic ions.

## 2.11. Chemical mass balance modeling

$PM_{2.5}$ OC was apportioned to its contributing sources using the EPA-CMB model (version 8.2)

using molecular marker concentrations in ambient $PM_{2.5}$ and source profiles as model inputs. Source profiles for garbage burning (fire #14A and 14B), open biomass burning (fire #39), biomass and dung powered traditional cooking stoves (fire #37, 38, 40, and 41), were drawn from NAMaSTE in 2015 (Jayarathne et al., 2018). Other primary and secondary source profiles were drawn from literature: vegetative detritus (Rogge et al., 1993), non-catalyzed gasoline

engines (Lough et al., 2007; Schauer et al., 2002), diesel engines (Lough et al., 2007), small-scale coal combustion (Zhang et al., 2008), isoprene, monoterpene, and sesquiterpene derived SOA (Kleindienst et al., 2007), and aromatic SOA from naphthalene and methyl-naphthalene (Kleindienst et al., 2012). The model sensitivity to the input source profiles were evaluated by systematically varying the biomass burning and garbage burning profiles developed in

NAMaSTE-2015 (Jayarathne et al., 2018), which examined the following source profiles: Open biomass burning, a mud stove fueled by wood and cow dung, a mud stove fueled by cow dung, a mud stove fueled by twigs, a mud stove fueled by wood, mixed garbage burning (samples A and B, discussed further in section 3.6).

## 2.12. Statistical analysis



Detectability of organic and inorganic species were 100%, except for 17α(H)-22,29,30-trisnorhopane (96%), 17β(H)-21α(H)-30-norhopane (96%), cholesterol (93%), stigmasterol (89%), 1-methylchrysene (93%), stigmastanol (67%), retene (56%), and coprostanol (30%). Prior to statistical analysis, data points with values below detection limits were replaced with the limit of detection (LOD)/√2 (Hewett and Ganser, 2007). All the concentrations were tested for normality and lognormality using the Anderson–Darling test. Concentrations of all the species were either normally or log-normally distributed, thus Pearson's correlation (r) was employed for correlation analysis. Two sample t-tests were used to compare the means of daytime and nighttime concentrations. All statistical tests were performed in Minitab (version 17) and significance was assessed at the 95% confidence interval ($p \leq 0.05$).

## 3. Results and discussion

### 3.1. $CO_2$, CO, $CH_4$, and NMVOCs

**3.1.1.** *VOC abundance.* Excluding oxygenated compounds, the ten most abundant NMVOCs in descending order based on median values were: ethene, ethyne, ethane, propene, propane, *i*-pentane, *i*-butane, *n*-butane, toluene and *m/p*-xylene. These and other selected NMVOC measurements in WAS samples are summarized in Table 2, with the corresponding precisions, accuracies, and detection limits.. Ethene and propene are major biomass burning products and major components of vehicle exhaust (Akagi et al., 2011; Guo et al., 2011), so their high abundance is expected given the prevalence of these sources. Ethyne is a general combustion tracer that is expected to reflect vehicular, biomass and biofuel combustion (Abad et al., 2011). Ethane also has multiple major sources including fossil fuel evaporation and combustion, biomass burning, and biofuel combustion (Guo et al., 2011; Xiao et al., 2008). Ethane correlated most strongly with the combustion tracer ethene (r = 0.81) and with $CH_4$ (r = 0.66), which has many anthropogenic sources, suggesting multiple ethane sources contribute. $CH_4$ is likely then to derive at least partially from combustion and is not expected to derive from natural gas production, processing, or transmission, due to a lack of natural gas infrastructure in the Kathmandu Valley in 2015, aside from a small number of household scale biogas plants. Additionally, the presence of numerous outliers in the data set suggests that individual grab samples were occasionally impacted disproportionately by local rather than regional sources (Table S1). The $C_3$-$C_5$ alkanes are not major biomass burning products, but are associated with





liquefied petroleum gas (LPG) and gasoline (Guo et al., 2011). Their abundance here suggests a traffic or fossil fuel source (as discussed in section 3.1.2). In urban centers toluene often reflects traffic, gasoline evaporation and/or solvents (Ou et al., 2015; Tsai et al., 2006). Four-stroke
engines are abundant in the Kathmandu Valley and their emissions are rich in aromatic VOCs (Shrestha et al., 2013). Here toluene correlated best with C4-C5 alkanes and ethylbenzene (r = 0.86 to 0.94) though surprisingly poorly with the vehicle exhaust tracer ethene (r = 0.01). While recognizing the limited sample size, this could suggest gasoline evaporation due to toluene's correlation with $i$-pentane (Tsai et al., 2006). Further study would help to clarify toluene's
sources.

Relative to previous measurements of NMVOC with a PTR-TOF-MS (proton transfer reaction time of flight mass spectrometer) at Bode from December 2012 to January 2013 during SusKat-ABC campaign (Sarkar et al., 2016), concentrations of the NMVOCs listed in Table 3 were generally lower during NAMaSTE in April 2015. Seasonal variability in meteorology
likely contributes to these differences. Mixing layer depths and associated dilution of regional emissions peak during the pre-monsoon season (March-May, including this study) whereas mixing layer depths are shallower during winter (Mues et al., 2017). Several rain events occurred during April 2015 (specifically on April 12 to 13, 15, 17 to 18, and 21) with a total of 24.2 mm of precipitation. Associated scavenging would have also contributed to the lower
pollution levels during this study relative to the dry winter season characterized during SusKat-ABC campaign. Notably, isoprene levels were nearly 10 times lower during April 2015 compared to the winter of 2012-13, suggesting lower contributions of biogenic VOCs. By April, nearly all deciduous trees in the Kathmandu Valley have leaves, although spring 2015 was unusually cold and the low temperatures leading up to and during the measurement period did
not favor isoprene emissions as further discussed in section 3.2.6. Additionally, in April 2015, VOC samples were not collected at mid-day when isoprene levels are expected to be highest. The whole air sampling in this study provides additional chemical detail to the high-time resolution measurements by proton transfer mass spectrometry (PTR-TOF-MS) during the SusKat-ABC intensive campaign (Sarkar et al., 2016), including resolution of alkane and
aromatic VOC isomers, and chlorofluorocarbons (Table S1).

In comparison to other cities in South Asia (Table 3), the NMVOC levels observed in Kathmandu were 1.3 to 8.5 times lower than in Mohali, India (Sinha et al., 2014), 2.6 to 6.7





times lower than Karachi, Pakistan (Barletta et al., 2002), 9.5 to 33 times lower than heavily
polluted Lahore, Pakistan (Barletta et al., 2017), and about a factor of two higher than in
Singapore (Barletta et al., 2017). Bearing in mind the small sample size, the observed NMVOC
levels in Kathmandu are reasonable compared to other studies and indicate that in April 2015 it
was moderately polluted with respect to other South and Southeast Asian cities, with relatively
low biogenic VOC influences.

**3.1.2.  *VOC sources.*** Because the NMVOC dataset is too small (n = 9) to be used in source
apportionment techniques like Positive Matrix Factorization (PMF), we use VOC ratios to
further probe source influences. All NMVOC ratios cited herein have been reported previously
by Simpson et al. (2014) and references therein, Akagi et al. (2011), and Stockwell et al. (2016).
The ratio of *i*-pentane/*n*-pentane increases from ~1 for natural gas to ~4 for gasoline evaporation.
Here, the ratio was 4.7 ± 0.4 (r = 0.97), consistent with gasoline evaporation as has been seen in
other cities such as Mecca, Saudi Arabia (Simpson et al., 2014). The diurnal ambient temperature
during this study ranged from 12 to 29°C, which is conducive to evaporation. As in Mecca, fuel
pump hoses in the Kathmandu Valley are not equipped with vapor recovery technology and
vehicles are not equipped with catalytic converters, which may partially explain the abundance
of the gasoline evaporation tracer *i*-pentane. Ethene/ethyne can be used to differentiate
petrochemical sources (10-30) from biomass burning (2-5) and vehicle exhaust (1-3), with ratios
below 1 reflecting older emissions control technology due to higher ethyne emissions. Here the
ratio was 0.5, similar to Saudi Arabia (0.73), suggesting a very large impact of older or poorly
maintained vehicles (Zhang et al., 1995). The *i*-butane/*n*-butane ratio can sometimes help to
distinguish influence from vehicles (~0.2 to 0.3), LPG combustion (~0.42 to 0.46) and natural
gas leaks (~0.6 to 1.0). Here the butane ratio was relatively high (~1, $r^2$ = 0.90) as compared to
cities in Saudi Arabia and Pakistan (0.4 to 0.6; Barletta et al. (2017); Simpson et al. (2014)). The
cause of the relatively abundant *i*-butane could be a mix of sources that needs further
investigation. Acetaldehyde was a major NMVOC consistent with past work (Table 3) and it has
a variety of poorly-constrained primary and secondary sources (Akagi et al., 2011; Stockwell et
al., 2016). These observations are consistent with the PMF analysis of NMVOC measurements
during the SusKat-ABC intensive campaign, which indicated that traffic and industrial emissions
were the largest sectors contributing to NMVOC mass loadings, at 17% and 18%, respectively
(Sarkar et al., 2017). The large diversity of combustion emissions in Kathmandu, the apparent





influence of point NMVOC sources, and chemical signatures not previously observed in South
Asia (discussed above) indicate that additional research with a larger sampling size is needed to
better understand NMVOC sources in the Kathmandu Valley.

### 3.2. Particulate matter and inorganic trace gases

### 3.2.1.  PM$_{2.5}$ and PM$_{10}$ concentrations

PM$_{2.5}$ mass concentrations averaged over 11 hours at the Bode supersite from 11 to 24 April
2015 ranged 30.0 to 207.4 µg m$^{-3}$ (Fig. 1), and averaged (±standard deviation) 68.2 ± 34.7 µg m$^{-3}$. PM$_{10}$ mass concentrations ranged 51.9 to 294.0 µg m$^{-3}$, and averaged 119.7 ± 55.2 µg m$^{-3}$. All
of the 11-hour PM$_{2.5}$ and PM$_{10}$ concentrations exceed the World Health Organization (WHO) 24-
hour guidelines of 25 µg m$^{-3}$ and 50 µg m$^{-3}$, respectively. The maximum concentrations of PM$_{2.5}$
and PM$_{10}$ occurred during the night of 11 April (Fig. 1), concurrent with the Bisket Jatra festival;
see Section 3.4 for a detailed description of this pollution event and its source characteristics.

The average PM$_{2.5}$ concentration observed in this study is nearly half of the mean of  24 hour
average PM$_{2.5}$ concentrations near 6 major road intersections in the Kathmandu Valley (125 µg
m$^{-3}$) during relatively drier period, February – April 2014 (Shakya et al., 2017b). The average
PM$_{2.5}$ concentration in the Kathmandu Valley was about a factor of two higher than at a more
rural and cleaner foothills site at Godavari (34 µg m$^{-3}$) during April 2006 (Stone et al., 2010),
and about a factor of 13 higher than the PM$_1$ concentration (5.4 µg m$^{-3}$) at the Nepal Climate
Observatory-Pyramid (NCO-P) site (near the basecamp for Mt. Everest) in the southern
Himalaya (27.95 N, 86.82 E) during March-April 2006 (Bonasoni et al., 2008). The average
PM$_{10}$ concentration in the Kathmandu Valley observed in this study period is similar to the
average annual concentration (155 ± 124 µg m$^{-3}$) of total suspended particles at the same site
between April 2013 and March 2014 (Chen et al., 2015).

PM concentrations at Bode were consistently higher during nighttime (83 µg m$^{-3}$ for PM$_{2.5}$
and 121 µg m$^{-3}$ for PM$_{10}$) compared to daytime (54 µg m$^{-3}$ for PM$_{2.5}$ and 117 µg m$^{-3}$ for PM$_{10}$).
The high-time resolution data for the AMS total signal (Fig. 2) indicate that PM mass increases
overnight, peaks around 08:00 local time, and thereafter decreases to minima around 17:00.
Diurnal variability in PM loadings is attributed to four interrelated factors. (1) Boundary layers
and the corresponding volumes of air into which pollutants are emitted are relatively shallower at
colder nocturnal temperatures (Mues et al., 2017). Although vertical temperature profiles were



not measured, the Kathmandu Valley frequently experiences shallow nocturnal inversion as evident in Ceilometer measurements during SusKat-ABC campaign (Mues et al., 2017). (2) Wind speeds during the pre-monsoon season and the corresponding dilution of PM emitted into or produced within that air flow are typically lower at night ($<1$ m s$^{-1}$) relative to daytime (1 to 5 m s$^{-1}$) (Fig. 2). The afternoon increase in wind speed corresponds to minimum PM values, while

lower wind speeds in early evening coincide with higher concentrations. (3) The diurnal wind dynamics in the Kathmandu Valley have been previously described (Mahata et al., 2017; Panday and Prinn, 2009; Panday et al., 2009; Sarkar et al., 2016). From midday to dusk, strong westerly flows carry pollutants from Kathmandu and Lalitpur towards the east and south passes of the Valley and the mixing layer height reaches its maximum. During evening, relatively stagnant

cooler air causes pollutants from the Bhaktapur Industrial Estate (which includes ~19 biomass- and coal-fired brick kilns) located within 1 to 5 km of Bode to accumulate near the surface, with slight elevation due to mild down-slope flows. In the early morning, elevated pollutants briefly recirculate back to the surface. Later in the morning, up-slope flows loft polluted air prior to the emergence of the strong westerly winds. (4) As discussed in more detail below, diel variability in

temperature and RH drove corresponding diel variability in aerosol liquid water content, aerosol solution pH, and the gas-aerosol phase partitioning of compounds with pH-dependent solubilities. The higher RHs and aerosol liquid water contents at night shifted partitioning towards the particulate phase thereby contributing to relatively higher PM mass concentrations at night. All of the above factors contribute to the relatively higher PM concentrations in near-

surface air at Bode during nighttime.

### 3.2.2. PM$_{2.5}$ organic and elemental carbon

OC concentrations ranged from 7.9 to 57.3 µg C m$^{-3}$ (averaging 17.6 ± 9.6 µg C m$^{-3}$) and accounted for 26 ± 5% of total PM$_{2.5}$ mass. The corresponding mass concentrations of organic

matter (OM) were estimated by multiplying OC mass concentrations by a factor of 1.7 to account for the associated elements (primarily oxygen, hydrogen, and nitrogen). The OM:OC conversion factor of 1.7 was obtained from the AMS measurement and falls towards the urban end of the range (1.6 – 2.1) recommended by Turpin and Lim (2001). OM accounted for an average of 48 ± 9% of PM$_{2.5}$ mass. EC concentrations ranged from 2.3-30.8 µg m$^{-3}$ (averaging 9.0 ± 6.4 µg m$^{-3}$)





and accounted for $13 \pm 6\%$ of $PM_{2.5}$ mass. Major sources for OC and EC are discussed in section 3.3.

### 3.2.3. Inorganic ions in PM, trace gases, and gas-aerosol phase partitioning

The major ionic components of PM were sulfate, ammonium, and nitrate accounting for $16 \pm$
$4\%$, $9 \pm 3\%$, and $4 \pm 2$ of $PM_{2.5}$, and $11 \pm 3\%$, $6 \pm 2\%$, and $3 \pm 2\%$ of $PM_{10}$, respectively (Table S2). Ratios of these ions indicate that secondary inorganic compounds including ammonium sulfate and ammonium nitrate were important components of PM (Carrico et al., 2003). The relative abundances of these species in the Kathmandu Valley are within the ranges of those reported previously (Shakya et al., 2010; Shakya et al., 2017a; Shakya et al., 2008).

Large diel variability in temperature and RH drove corresponding variability in aerosol LWC, which contributed to diel variability in the phase partitioning of $NH_3/NH_4^+$, $HNO_3/NO_3^-$, and $HCl/Cl^-$ and aerosol solution pH (Fig. 3). In contrast, under acidic conditions (as existed during this campaign, see below) and in the presence of high aerosol surface area, $SO_2$ and $H_2SO_4$ are relatively insensitive to variability of aerosol LWC and solution pH. Under these
conditions, virtually all $SO_2$ partitions into the gas phase and virtually all $H_2SO_4$ partitions into the particulate phase. Consequently, the phase partitioning of oxidized S (Fig. 3f) can be interpreted without complications introduced by corresponding phase changes in response to variable LWC and Ph.

Both $SO_2$ and particulate $SO_4^{2-}$ were systematically higher at night and lower during the
daytime (Table 4, Fig. 3f). Average concentration of particulate $SO_4^{2-}$ measured with the AMS followed a similar day-night trend, with peak concentrations occurring around 08:00 local time (Fig. 2). The corresponding total oxidized S ($SO_2$ + particulate $SO_4^{2-}$) during daytime versus nighttime (Table 4) typically differed by factors of 2 to 5. If photochemical oxidation of $SO_2$ to $H_2SO_4$ had contributed significantly to the diel variability in $SO_2$, $SO_2$ and particulate $SO_4^{2-}$
would have been anti-correlated, which was not the case. These results imply that diel variability in atmospheric dynamics (wind velocity, boundary layer depth, and transport of chemically distinct air masses within the valley) were major drivers of the observed variability in both species as discussed in Section 3.2.1.

Total $NH_3$ ($NH_3$ + particulate $NH_4^+$) exhibited a diel pattern similar to that of oxidized S
(Table 4, Fig. 3c) although relative day-night differences were proportionally smaller (typically



less than a factor of 2). Day-night differences in total $NO_3$ ($HNO_3$ + particulate $NO_3^-$) and total Cl ($HCl_{calc}$ + particulate $Cl^-$) (Table 4, Fig. 3d and 3e, respectively) were somewhat more variable but also tended to be higher at night than during the day. Taken together, the above results support the hypothesis that transport of chemically distinct air masses from different

source regions during daytime versus nighttime was a major factor that drove diel variability in the composition of the multiphase gas-aerosol system at Bode. Concentrations of $Na^+$ (in nmol $m^{-3}$) associated with $PM_{2.5}$ (median – 4.0, range – undetectable to 19.0) and $PM_{10}$ (median – 8.7, range – undetectable to 40.9) were typically much lower than those of particulate $Cl^-$ and total Cl (Fig. 3e). These relationships indicate that, in contrast to some other continental regions (e.g.

Young et al., 2013, Jordan et al., 2016), refractory NaCl emitted from crustal and/or marine sources was not the primary source for particulate and volatile Cl at Bode. Instead, total Cl ($HCl_{calc}$ + particulate $Cl^-$) showed high correlation with potassium (r = 0.91, p <0.001) and total $NH_3$ ($NH_3$ + particulate $NH_4^+$) (r = 0.78, p < 0.001), suggesting their co-emission from biomass burning (Jayarathne et al., 2018; Keene et al., 2006; Sheesley et al., 2003), brick kilns (Stockwell

et al., 2016), and/or garbage burning (Jayarathne et al., 2018) in the Kathmandu Valley.

Concentrations of volatile inorganic Br (including HBr, $Br_2$, HOBr, BrCl, and BrO) ranged from <0.54 to 1.18 nmol $m^{-3}$ and were greater than the detection limit in 8 of 27 samples (Fig. 3g). Seven of the 8 detectable mixing ratios were during nighttime sampling intervals, which suggest a possible diel cycle in multiphase chemical processing of Br and/or systematic

variability as a function of transport from different source regions. $Br^-$ was not measured in aerosol samples so corresponding variability of particulate and total (volatile + particulate) Br is not known. Possible sources for reactive Br in the region include biomass burning and fossil-fuel combustion (Sander et al., 2003).

Concentrations of volatile and particulate inorganic Cl measured at Bode fell within the

ranges of those measured in polluted continental air (Young et al., 2013). In addition, concentrations of volatile inorganic Br and Cl at Bode fell within the ranges of those measured in marine air (Keene et al., 2009; Sander et al., 2003). Drawing on related model calculations and measurements elsewhere (Keene et al., 1999; Keene et al., 2009; Long et al., 2014; Sander et al., 2003; Young et al., 2013), these results in conjunction with the presence of acidic, deliquesced

aerosol support the hypothesis that multiphase halogen-radical chemistry involving both Br and Cl impacted regional air quality in the Kathmandu Valley during the campaign via two





pathways. (1) At high $NO_x$ mixing ratios in polluted continental regions, the nocturnal reaction of $N_2O_5$ with particulate $Cl^-$ produces significant $ClNO_2$, which photolyzes following sunrise yielding a burst of Cl atoms (e.g., Brown et al. (2013)). $ClNO_2$ is also a nocturnal reservoir for

$NO_x$ and thereby slows $NO_x$ destruction at night. (2) The scavenging of volatile HOCl and HOBr into acidic aerosol solution and their subsequent reaction with $Cl^-$ and $Br^-$ produces $Cl_2$, BrCl, and $Br_2$, which subsequently volatilize to the gas phase and photolyze during daytime yielding atomic Br and additional atomic Cl (e.g., Keene et al. (2009)). These autocatalytic reactions proceed in both the light and dark and would enhance halogen activation at night and

sustain halogen radical chemistry during daytime relative to predictions based on $ClNO_2$ activation alone. The associated production and scavenging of halogen nitrates also accelerates the destruction of $NO_x$. Cl and Br radicals contribute to oxidation of hydrocarbons and, together with related reactions that impact $NO_x$ cycling, perturb $HO_x$-$NO_x$ photochemistry relative to that predicted in the absence of reactions involving halogens.

Particulate calcium, magnesium, and fluoride in the Himalayan region originate primarily from the deflation of surface soils (Carrico et al., 2003). Contributions of these ions to $PM_{10}$ mass were statistically greater (p=0.03, 0.01, and <0.001, respectively) relative to those to $PM_{2.5}$ (Table S2) consistent with results reported previously by Hinds (2012). Heavy vehicular traffic on the many unpaved roads in the valley, and to some extent wind-blown soil dust from the open

agriculture fields (by relatively stronger winds in pre-monsoon season) likely contributed to regional dust emissions. This speculation is supported by the significantly higher concentrations of the calcium and magnesium associated with $PM_{10}$ during daytime (p=0.049 and 0.005, respectively) when vehicular traffic is higher relative to night. The long-distance transport of dust from arid regions upwind may also contribute to that produced locally.

**3.2.4.  Aerosol pH**

Aerosol solution pH, inferred from the phase partitioning of $NH_3$ and $HNO_3$, ranged from 2.2 to 3.3 and most paired estimates for the same sampling interval agreed within ±0.1 to ±0.3 pH units (Fig. 3h). Solution pHs estimated from the partitioning of a given gas with $PM_{10}$ were greater than those for $PM_{2.5}$ by <0.01 to about 0.3 pH units. These results indicate that particles

larger than 2.5-µm diameter were typically less acidic that than smaller particles, which is consistent with size-resolved acidities characterized in other continental regions (e.g., Young et al., 2013). These results are also consistent with the proportionately greater divergence between



concentrations of $NO_3^-$ (Fig. 3d) and $Cl^-$ (Fig. 3e) associated with $PM_{10}$ versus $PM_{2.5}$ relative to those for $NH_4^+$ (Fig. c).  Based on their thermodynamic properties, $HNO_3$ and $HCl$ partition

preferentially with less acidic particles whereas $NH_3$ partitions preferentially with the more highly acidic particles.

### 3.2.5.  PM$_{2.5}$ molecular markers and secondary organic aerosol tracers

Organic species, particularly molecular markers and SOA tracers (Table 1), were measured to

identify sources of OC (Table 5). The identified sources include biomass burning, food cooking, dung burning, garbage burning, coal combustion, fossil fuel use, and secondary aerosol from both biogenic and anthropogenic sources.

Herein, we present the first measurements of 1,3,5-triphenylbenzene (TPB) in Nepal, which is unique molecular marker of plastic burning. This tracer is associated with the

combustion of garbage (Jayarathne et al., 2018; Simoneit et al., 2005) which occurs widely in Nepal (Wiedinmyer et al., 2014). Open garbage/trash burning is recognized as a common method to dispose of waste materials in Nepal and other South Asian countries (Wiedinmyer et al., 2014). Garbage contains plastic items (e.g., bags, packaging, food containers) in addition to food waste, paper, cardboard, foil packaging, and other items (Stockwell et al., 2016). TPB was

detected in every sample (Fig. 4a). Ratios of TPB to OC concentrations during daytime and nighttime ($0.04 \pm 0.02$ ng µGoc$^{-1}$) suggests that garbage burning contributed generally consistent fractions of $PM_{2.5}$ OC. In addition, TPB was significantly correlated with the biomass burning marker levoglucosan ($r = 0.66$, $p < 0.001$), which suggests that both tracers may have been emitted during the co-combustion of plastic materials and biomass including food waste, paper,

and cardboards in garbage.

Additional molecular markers indicated the presence of biomass burning, coal combustion, fossil fuel use, dung burning, and food cooking. Levoglucosan, a biomass burning marker, was observed throughout the sampling period (Fig. 4b). Biomass burning emissions that impact  the Kathmandu Valley during April include local biomass combustion in brick kilns and

food cooking (Pariyar et al., 2013), as well as garden waste burning, agro-residue burning, and regional wild fires (Khanal, 2015; Stone et al., 2010). Picene, a marker of coal combustion (Oros and Simoneit, 2000) and tire burning (Downard et al., 2015) was consistently detected (Fig. 4c). Picene varied diurnally, with higher concentrations at nighttime ($0.25 \pm 0.13$ ng µgOC$^{-1}$)



compared to daytime ($0.14 \pm 0.16$ ng $\mu$gOC$^{-1}$), consistent with southeasterly winds at night that transported emissions from coal-fired brick kilns to Bode. Nine hopanes were identified, indicating fossil fuel influences on PM$_{2.5}$ in the form of coal burning and/or vehicle emissions (Fig. 4d). Stigmastanol, a unique molecular marker of cow dung burning (Sheesley et al., 2003) was detected in about 67% of the samples (Fig. 4e, Table 5). Cholesterol, which is emitted from both dung burning (Sheesley et al., 2003) as well as food cooking (Rogge et al., 1991), was detected in 93% samples (Fig. 4f). Cholesterol and stigmastanol exhibited distinct temporal variabilities, the former having higher OC-normalized concentrations during nighttime and the latter having higher OC-normalized concentrations during daytime. This suggests that these two were emitted from different sources, i.e. cholesterol from food cooking.

Concentrations of molecular markers in Bode measured in this study are compared to a more rural site at Godavari, in the outskirts of the Kathmandu Valley, ca. 10 km south of Bode Nepal, during April 2006 (Stone et al., 2010). In contrast to Bode, Godavari is located ~11 km south of Bode at the base of the western face of a mountain (Phulchoki) that rises ~1200 m above the valley floor, with very low population density nearby. While Bode experiences afternoon westerlies that cross the valley from the western to the eastern passes, the flow in Godavari is dominated by the up and downslope flows generated by Mt. Phulchoki. Higher concentrations of most markers at Bode, including levoglucosan (by a factor of 5), picene (by a factor of 23), 17$\beta$(H)-21$\alpha$(H)-30-norhopane (by a factor of 13) indicate the larger impact of biomass burning and fossil fuel combustion at this site, compared to Godavari, likely due to the higher population density and industrial activities. The average concentration for stigmastanol at Bode (1.02 ng m$^{-3}$) was similar to Godavari (0.9 ng m$^{-3}$) (Stone et al., 2010). Although dung burning is not common in the Kathmandu Valley and its outskirts, dung is a more widely used fuel in rural areas of southern Nepal and India and may contribute to the observed dung burning tracers.

SOA produced from the oxidation of the biogenic precursors (monoterpenes and sesquiterpenes) were indicated by *cis*-pinonic acid and $\beta$-caryophyllinic acid, respectively (Fig. 4g, Table 5). OC normalized concentrations of *cis*-pinonic acid (ratio of *cis*-pinonic acid to total OC associated with PM$_{2.5}$) were significantly higher during daytime compared to nighttime ($p < 0.001$), which would be consistent with the photochemical production of *cis*-pinonic acid during daytime (Claeys et al., 2007; Kleindienst et al., 2007) but may also reflect relative differences in emissions of precursors for *cis*-pinonic acid in air mass source regions upwind of





the site during daytime versus nighttime. In contrast, isoprene concentrations during the study period were low (Table 2) and corresponding concentrations of methyltetrols (tracers of SOA produced from the oxidation of isoprene) were below detection limits. These results are consistent with expectation of low isoprene emissions from recently emergent leaves coupled with relatively low temperatures as discussed above (Kuzma and Fall, 1993; Lewandowski et al.,

2008; Monson et al., 1992; Shen et al., 2015).

      Phthalic acid and 4-methylphthalic acid (Table 5) are photo-oxidation products of naphthalene and methyl-naphthalene, respectively (Kleindienst et al., 2012). These species were also reported to be observed from vehicle emission (Kawamura and Kaplan, 1987), but may be used as anthropogenic SOA tracers in an absence of correlation with primary source tracers (e.g.,

hopane) (Al-Naiema and Stone, 2017). Phthalic acid and 4-methylphalic acid did not correlate significantly with hopanes (r=0.29 and 0.25, respectively) and EC (r=0.16 and 0.12, respectively), suggesting that primary combustion was not their major source. Both species were present significantly higher OC normalized concentrations during the daytime (p<0.001). The daytime maxima may be due to photochemical production (Kleindienst et al., 2012) and/or

transport of air masses that passed over Kathmandu during the daytime.

### 3.3. Chemical Mass Balance Source Apportionment Modeling of $PM_{2.5}$ OC

### 3.3.1. Base case model result

Chemical mass balance (CMB) modeling was used to apportion $PM_{2.5}$ OC to five primary

sources (garbage burning, biomass burning inclusive of open burning and biomass fueled cooking stoves, gasoline and diesel engines, coal combustion, and vegetative detritus) and four secondary sources (monoterpene SOA, sesquiterpene SOA, naphthalene SOA, and methylnaphthalene SOA). The "base case" results represent the best estimate of the source contributions to OC in this study and utilizes the most representative source profiles available

(see section 2.11). Of those that were resolvable, primary sources contributed an average of 60 ± 16% of OC, secondary sources accounted for 15 ± 5% of OC, while the remaining 25 ± 16% of OC was not apportioned and is referred to as "other sources" (Fig. 5, Table 6). Other sources may include contributions from cooking with non-biomass fuels (e.g., LPG), mixed industrial emissions, dust emissions, and other uncharacterized primary and secondary sources that could

not be apportioned because marker species were not measured (e.g., Si, Al in the case of dust),



source profiles are not available (e.g., for local industry), or available profiles are considered unsuitable (e.g., for cooking activities, given the inherent variability of this source). Other sources may also include OC from apportioned sources, in the case that they were underestimated. The CMB model did not converge on April 13-N, and 18-D, and thus primary

source contributions are not reported for these samples.

The garbage burning contribution to $PM_{2.5}$ OC ranged from 11 to 27% and averaged 18 ± 4%. These results indicate that garbage burning to OC is a major source of $PM_{2.5}$ in the Kathmandu Valley. To our knowledge, this is the first study to apportion $PM_{2.5}$ OC to garbage burning source based on a unique molecular marker (TPB) in CMB. A tracer based estimation of

garbage burning contributions to $PM_{2.5}$ in the Mexico City Metropolitan Area using antimony (Sb) as a tracer indicated that garbage burning contributed ~28% of $PM_{2.5}$ (Christian et al., 2010). Hodzic et al. (2012) estimated that organic aerosols in the Mexico City Valley could be reduced by 2 to 40% by complete mitigation of garbage burning. The large estimates of garbage burning contributions to $PM_{2.5}$ OC and $PM_{2.5}$ in these Kathmandu and Mexico City demonstrates

the importance of this source to local air quality in heavily polluted urban air. Further, this source should be considered source apportionment in regions where open garbage burning is a common practice.

Biomass burning contributed 5 to 43% of $PM_{2.5}$ OC, averaging 17 ± 10%. This estimate is expected to encompass a wide range of biomass burning sources, including biofuel and open

burning of biomass. Biomass is widely used as a fuel for household cooking and heating (Pattanayak et al., 2005; Pokhrel et al., 2015; Yevich and Logan, 2003) and in brick kilns (Maithel et al., 2012; Stockwell et al., 2016). In addition, burning of agricultural residue and wild fires commonly occur in April in Nepal (Khanal, 2015). The source profile used for biomass burning in Nepal was based on characterization of emissions from an open biomass fire of twigs

and dung (Jayarathne et al., 2018) to reflect that molecular markers for wood burning (levoglucosan) and dung burning (stigmastanol) were present. Vegetative detritus, which is waxy material produced from abrasion of plant leaves contributed an average of 1.6 ± 0.9% of $PM_{2.5}$ OC and is sometimes associated with biomass burning emissions, due to lofted vegetative matter during combustion. A small contribution from this source is consistent with prior studies in the

region (Stone et al., 2010).



Contributions from fossil fuel combustion to $PM_{2.5}$ OC included emissions from gasoline and diesel engines and coal combustion. Greater than 1 million vehicles (Mahata et al., 2017) and approximately 0.25 million power generators (World Bank, 2014) fueled by gasoline and diesel operate within the Valley. The combined OC contributions to $PM_{2.5}$ OC from these

sources ranged 5 to 48% and averaged 18 ± 9% (Table 6). These two sources are reported together because they both contribute to evaporative emissions of motor oil. Coal combustion contributed 1 to 10% and averaged 5.0 ± 2.3% of to $PM_{2.5}$ OC. Coal combustion contributions to $PM_{2.5}$ OC at Bode were significantly greater during nighttime periods (5.9 ± 2.3%) compared to daytime (3.9 ± 2.0%, p=0.04). As discussed above, these diel differences reflect the proximity of

coal fired brick kilns located to the south and east of the sampling site coupled with the transport of emissions via the south-easterly winds at night. Relative contributions of coal combustion to $PM_{2.5}$ (0.8 + 0.4 µgC m$^{-3}$) in the Kathmandu Valley are about four times greater than those at a more rural local in Nepal (Stone et al., 2010). Brick kilns operate only in the dry season in Nepal and not during the summer monsoon (rainy) season, making this a seasonal $PM_{2.5}$ source.

Naphthalene-derived SOC was the largest identified SOA source, contributing 10 ± 4% of OC, while methylnaphthalene-derived SOC contributed 0.3 ± 0.1%. Average contributions of biogenic SOC to $PM_{2.5}$ OC ranged from 0.03 to 0.29% for monoterpenes and 1.5 to 8.3% for sesquiterpenes. As noted above, isoprene SOA tracers were not detected. Relatively lower contributions from biogenic SOC during winter and post-winter months were reported in

previous studies in Nepal (Stone et al., 2010), Southeastern US (Kleindienst et al., 2007), Midwestern US (Lewandowski et al., 2008). Thus, the SOA in the Kathmandu Valley during pre-monsoon season was dominated by anthropogenic influences with ~70% contribution to total SOC. Naphthalene and methylnaphthalene were reported to mainly come from diesel exhaust (Schauer et al., 1999), and industrial emissions and biomass burning (Jia and Batterman, 2010),

indicating that reduction of emissions from these sources would reduce anthropogenic SOA production. It is likely that other SOA sources (e.g., those associated with monoaromatic VOCs, biomass burning, and other SOA precursors) that were not characterized in this study also contributed to $PM_{2.5}$ OC. Contributions from all four SOA sources to $PM_{2.5}$ OC were significantly greater (p<0.001) during daytime (18 ± 4%) relative to nighttime (11 ± 3%) (Table

5). These results are consistent with those reported in other studies (Kleindienst et al., 2007; Plewka et al., 2006; Xu et al., 2015) and are attributed to photochemical production from





precursors during daytime and/or regional variability in emissions of precursors in upwind source regions for air transport to Bode during daytime versus nighttime. The low source contribution of biogenic relative to anthropogenic SOC is consistent with the low biogenic VOC

levels (Table 2). Based on results from the SusKat-ABC campaign, oxidation products of aromatic VOCs (most importantly benzene) probably also contributed substantially to SOC (Sarkar et al., 2017). In this study, the uncharacterized contributions from aromatics would be classified as other/unapportioned OC.

**3.3.2.   Sensitivity to garbage and biomass burning profiles**
The sensitivity of the CMB source apportionment results to the input source profiles was examined by systematically varying either the garbage burning or biomass burning source profiles while keeping other profiles constant, following prior studies (Sheesley et al., 2003; Stone et al., 2010). The sensitivity test results are summarized in Fig. 6, and the model

performance metrics are summarized in Fig. S1.

Two garbage burning profiles were examined from a single fire of mixed waste burning (NAMaSTE fire #14; (Jayarathne et al., 2018)). Profile A (the base case profile) corresponds to more smoldering conditions with modified combustion efficiency (MCE) of 0.89 and profile B corresponding to a mixture of flaming and smoldering combustion with and MCE of 0.93

(Jayarathne et al., 2018). The mixed waste included food waste, paper, plastic bags, cloth, diapers, and rubber shoes. These garbage materials were damp with previous night's rainfall and were rekindled with newspaper on occasions (Stockwell et al., 2016). Switching from profile A to B increased the amount of $PM_{2.5}$ OC apportioned to garbage burning by a factor of 3.0 (Fig. 6a) to 46 ± 13% of OC, indicating that the model was highly sensitive to the garbage burning

profile. The model result from profile B, on some days, caused the CMB apportioned OC to exceed the observed OC. Garbage tends to be burned inefficiently and it is possible that this source contributes to more OC estimated in the base-case scenario and may account for much of the unapportioned OC. Given its potential contributions to $PM_{2.5}$, additional sampling and characterization of this source is warranted.

Burning of wood, dung, and crop residue are major energy sources of households in South Asia, while crop residue is also burned in in fields (Saud et al., 2011; Yevich and Logan, 2003). An open biomass fire of twigs and dung was used as the base case biomass profile



(NAMaSTE fire #39) (Jayarathne et al., 2018). Four other biomass burning profiles 1-pot traditional mud cooking stove fueled with: hardwood (fire #37), twigs (fire #38), dung (fire #40),
hardwood & dung (fire #41) were examined and the OC apportioned to biomass burning changed by factors of 0.4–1.9 (Fig. 6b). The lowest estimate corresponded to wood and dung fueled mud cooking-stove and highest estimate corresponded to a wood-fueled mud cooking-stove, with 15 ± 8% and 39 ± 16% OC apportioned to biomass burning (primarily cooking), respectively. The agreement of the sensitivity tests to the base case results being within a factor of two indicates a
relatively stable CMB-apportionment of biomass burning to OC. Because some of the sensitivity tests predict a higher biomass burning contribution to OC than the base case result, this source may contribute to some of the unapportioned OC in the base case result.

The base case CMB model apportioned $PM_{2.5}$ EC to five primary sources (garbage burning, biomass burning, gasoline and diesel engines, coal combustion, and vegetative detritus).
The largest contributor of EC was gasoline and diesel engines (89 ± 7%), with smaller contributions from biomass burning (7 ± 7%), coal combustion (3 ± 2%), garbage burning (1 ± 1%) and vegetative detritus (0.1 ± 0.1%) (Fig. S2). EC apportionment was highly sensitive to different biomass and garbage profiles (Fig. S3) with the EC apportioned to garbage burning varying by a factor of 10 and EC apportioned to biomass burning varying by a factor of 0.2-3.4.
Due to the large model sensitivity to the selected profile, the EC apportionment is not considered to be as robust as the OC apportionment and is subject to larger uncertainties. To better constrain EC source contributions, additional measurements (i.e. radiocarbon) would be needed.

### 3.4. PM composition and sources during Bisket Jatra, the local New Year festival

A 9-day (10-18 April) festival, Bisket Jatra, celebrating the start of the local New Year in the Bikram Sambat calendar began on 14 April 2015in Bhaktapur located in the southeast corner of the Kathmandu Valley. Two main events occurred on the afternoons and evenings of 11 and 13 of April that attracted a large number of spectators, increased vehicle traffic in the surrounding area, and involved cooking food both indoors and outdoors food. The prevailing southeasterly
winds during nighttime would have transported air masses from Bhaktapur to Bode. The maximum concentrations of $PM_{2.5}$ and $PM_{10}$ measured over the campaign (207 µg m$^{-3}$ and 294 µg m$^{-3}$, respectively) were during night of 11 April (Fig. 1). The $PM_{10}$ concentration on the night of 13 April (145 µg m$^{-3}$) was also relatively high compared to other sampling periods; the




corresponding PM$_{2.5}$ concentration was not available due to filter damage. OC and EC concentrations were about three times higher on 11 April and two times higher on 13 April compared to the average values over the study period. Compared to more typical concentrations based on average values during the study period, on 11 April, levoglucosan was three times greater and on 13 April, levoglucosan, cholesterol, and hopanes were five, eleven, and five to six times greater. The increased emissions from cooking and vehicle traffic associated with the festival were the most likely sources for the relatively higher concentrations of these molecular markers.

The base case CMB source apportionment on the night of 11 April indicated that sources contributing to PM$_{2.5}$ OC were biomass burning (26 ± 8%), gasoline and diesel engines (19 ± 3%), garbage burning (12 ± 4%), coal combustion (1.6 ± 0.5%), vegetative detritus (1.5 ± 0.8%), monoterpene SOC (0.03 ± 0.02%), sesquiterpene SOC (3.1 ± 4.2%), naphthalene SOC (3.7 ± 0.8%), methylnaphthalene SOC (0.09 ± 0.12%), and other sources (34 ± 15%). The biomass burning contribution to PM$_{2.5}$ OC on the night of 11 April was approximately 1.5 times higher than the average biomass burning contribution during the study period. The magnitude of biomass burning and gasoline and diesel engine contributions to PM$_{2.5}$ OC on that night were the highest (15 ± 4 µgC m$^{-3}$ and 11 ± 1 µgC m$^{-3}$, respectively) among the study period, which were approximately 5 times and 4 times higher than the average contributions of these sources over the study period. The relatively higher contributions from these two sources on the night of 11 April indicated the influences of Bisket Jatra festival on the air quality in the Kathmandu Valley. In contrast, contributions of other primary sources (garbage burning, coal combustion, and vegetative detritus) and SOA to PM$_{2.5}$ OC were approximately 2 to 3 times lower on the night of 11 April compared to their study averages. The CMB model did not converge for the sample collected on the night of 13 April, likely because the local pollution sources on that night (e.g., meat cooking indicated by cholesterol) were not well represented by the source profiles in the model.

## 4. Conclusions

Filter sampling and off-line analyses showed that primary combustion sources were the major contributors to volatile and reactive gases, PM$_{2.5,}$ OC, and EC in Kathmandu in mid-April 2015 (pre-monsoon). Using regionally-specific source profiles when available, major primary




OC sources were estimated to be garbage burning (18 ± 5%), biomass burning (17 ± 10%), and
gasoline and diesel engines (18 ± 9%). This study provides the first apportionment of PM to
garbage burning in South Asia, and indicates that it is among the major OC sources. However,
the model sensitivity tests indicate that the garbage burning source contribution can vary widely
depending on the input source profile (and increase by up to a factor of three), indicating that this

source may have an even larger impact on PM. Garbage burning contributions to PM may be
further constrained with other elemental tracers (e.g., Sb) and garbage burning should be further
characterized in terms of its variability with respect to garbage composition and combustion
conditions. Since garbage burning, biomass burning, and vehicle emissions are, at least in part,
controllable and are thus potential targets for emissions reductions to reduce ambient $PM_{2.5}$ in the

Kathmandu Valley. Mitigation strategies could include improvements to waste management;
higher efficiency of biomass use to reduce PM emissions from cooking, heating, or brick kilns;
and reductions in emissions from vehicles.

This study characterized air quality in the Kathmandu Valley for thirteen days but was
halted prematurely by the Gorkha earthquake that struck Nepal on 25 April 2015. A longer

period of study is required to better understand the seasonal variation of pollution sources and
the role of SOA during periods of higher biogenic VOC levels. Approximately 30% of the $PM_{2.5}$
OC was un-apportioned to the sources evaluated in this model. Likely additional sources include
evaporative emissions from vehicles, local industries, agricultural burning, and SOA from
monoaromatic species (like benzene, toluene, etc.) that were not characterized in this study.


**Supplemental information**: Table S1: Concentrations of methane, CO, $CO_2$, COS, and select
non-methane volatile organic compound (NMVOC) measured at Bode in April 2015. Table S2:
Mean (± standard deviation) PM mass fractions (as %) of water-soluble inorganic ions; Table S3:
Ambient concentrations of $PM_{10}$ mass and inorganic ions measured at Bode in the Kathmandu

Valley; Figure S1: Comparison of CMB model performance metrics for the sensitivity tests
using different biomass and garbage burning profiles; Figure S2: Apportionment of primary and
secondary sources for $PM_{2.5}$ EC based on CMB modeling; Figure S3: Sensitivity of CMB model
results to the input source profiles.




**Author contributions:** EAS, WCK, PFD, RJY, MR, and AKP designed and directed the study;
TJ, BW, PSP, SA, AKP, MR, and PFD conducted field operations, collected field samples,
and/or collected field data; MRI, TJ, IJS, JM, AG, DRB, and WCK conducted laboratory
analyses of samples; MRI, TJ, IJS, BW, JM, AG, DRB, RJY, PFD, WCK, and EAS analyzed
data; MRI and EAS conducted CMB modeling. All authors contributed to writing and/or
reviewing the paper.

*Acknowledgements.* This project was funded by the National Science Foundation through the
grant entitled "Collaborative Research: Measurements of Selected Combustion Emissions in
Nepal and Bhutan Integrated with Source Apportionment and Chemical Transport Modeling for
South Asia via award numbers AGS-1351616 to the University of Iowa, AGS-0003865 to the
University of Virginia (UVA), and AGS-1349976 to the University of Montana, and with the
grant entitled "Ambient and Source Characterization of Aerosol Size and Composition in Nepal
and Bhutan using High-Resolution Aerosol Mass Spectrometry" via award number AGS-
1461458 to Drexel University. Maheswar Rupakheti was supported by the Institute for Advanced
Sustainability Studies (IASS) which is funded by the German Ministry of Education and
Research (BMBF) and the Brandenburg State Ministry of Science, Research and Culture
(MWFK). NAMaSTE was partially supported by core funds of ICIMOD contributed by the
governments of Afghanistan, Australia, Austria, Bangladesh, Bhutan, China, India, Myanmar,
Nepal, Norway, Pakistan, Switzerland, and the United Kingdom, as well as by funds from the
Government of Sweden to ICIMOD's Atmosphere Initiative. We also thank Pratik Man Singdan,
Bhogendra Kathayat, and Shyam Kumar Newar from Nepal for their help in sample collection.

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





**Table 1:** Molecular markers for primary or secondary sources of particulate matter.

| Source | Molecular marker | Reference |
|---|---|---|
| Biomass burning | Levoglucosan | (Simoneit et al., 1999) |
| Fossil fuel combustion/evaporation | Hopanes | (Schauer et al., 1999) |
| Food cooking | Sterols | (Rogge et al., 1991) |
| Cow dung burning | Stigmastanol | (Sheesley et al., 2003) |
| Garbage/plastic burning | 1,3,5-Triphenylbenzene | (Simoneit et al., 2005) |
| Vegetative detritus | *n*-Alkanes with odd carbon preference | (Rogge et al., 1993) |
| Isoprene SOA | Methyltetrols | (Kleindienst et al., 2007) |
| Monoterpene SOA | *cis*-Pinonic acid | (Kleindienst et al., 2007) |
| Sesquiterpene SOA | β-Caryophyllinic acid | (Jaoui et al., 2007) |
| Naphthalene SOA | Phthalic acid | (Kleindienst et al., 2012) |
| 2-Methylnaphthalene SOA | 4-Methylphthalic acid | (Kleindienst et al., 2012) |





**Table 2:** Means, standard deviations, medians, and ranges of concentrations of methane, CO, $CO_2$, COS, and select non-methane volatile organic compound (NMVOC) mixing ratios measured at Bode in April 2015 (n=9). Species reported here include the twenty most abundant species, with all NMVOC measurements are provided in Table S1. Units are ppbv, unless noted.


| Compound | Precision (%) | Accuracy (%) | Mean ± std. dev. | | | Median | Range | | |
|---|---|---|---|---|---|---|---|---|---|
| $CH_4$ (ppmv) | 0.1 | 1 | 1.999 | ± | 0.082 | 1.976 | 1.926 | - | 2.188 |
| CO (ppmv) | 2 | 5 | 0.766 | ± | 0.751 | 0.509 | 0.362 | - | 2.737 |
| $CO_2$ (ppmv) | 2 | 2 | 425 | ± | 8 | 424 | 415 | - | 435 |
| Carbonyl sulfide | 2 | 10 | 0.66 | ± | 0.20 | 0.59 | 0.47 | - | 1.13 |
| $CH_3Cl$ | 5 | 10 | 0.87 | ± | 0.22 | 0.86 | 0.67 | - | 1.42 |
| Ethane | 1 | 5 | 2.29 | ± | 0.39 | 2.20 | 1.69 | - | 2.74 |
| Ethene | 3 | 5 | 3.20 | ± | 1.11 | 3.26 | 1.59 | - | 4.64 |
| Ethyne | 3 | 5 | 3.06 | ± | 1.46 | 2.68 | 1.51 | - | 6.31 |
| Propane | 2 | 5 | 2.05 | ± | 1.69 | 1.39 | 0.69 | - | 5.77 |
| Propene | 3 | 5 | 1.92 | ± | 1.04 | 1.68 | 0.54 | - | 3.51 |
| *i*-Butane | 3 | 5 | 1.47 | ± | 0.97 | 1.29 | 0.28 | - | 3.21 |
| *n*-Butane | 3 | 5 | 1.39 | ± | 0.80 | 1.27 | 0.23 | - | 2.47 |
| *i*-Butene | 3 | 5 | 0.93 | ± | 0.80 | 0.82 | 0.12 | - | 2.81 |
| 1,3-Butadiene | 3 | 5 | 0.13 | ± | 0.09 | 0.17 | 0.02 | - | 0.28 |
| *i*-Pentane | 3 | 5 | 1.77 | ± | 1.76 | 1.38 | 0.10 | - | 6.07 |
| *n*-Pentane | 3 | 5 | 0.46 | ± | 0.37 | 0.45 | 0.03 | - | 1.25 |
| *n*-Hexane | 3 | 5 | 0.29 | ± | 0.26 | 0.21 | 0.03 | - | 0.80 |
| *n*-Heptane | 3 | 5 | 0.24 | ± | 0.36 | 0.10 | 0.02 | - | 1.18 |
| *n*-Octane | 3 | 5 | 0.12 | ± | 0.08 | 0.09 | 0.05 | - | 0.30 |
| *n*-Nonane | 3 | 5 | 0.18 | ± | 0.14 | 0.16 | 0.04 | - | 0.42 |
| Benzene | 3 | 5 | 1.01 | ± | 0.49 | 0.86 | 0.59 | - | 2.19 |
| Toluene | 3 | 5 | 0.99 | ± | 0.53 | 1.06 | 0.30 | - | 1.84 |
| *m/p*-Xylene | 3 | 5 | 1.11 | ± | 0.61 | 1.02 | 0.20 | - | 2.26 |
| *o*-Xylene | 3 | 5 | 0.46 | ± | 0.29 | 0.40 | 0.13 | - | 0.89 |
| Methanol | 30 | 20 | 4.38 | ± | 1.66 | 3.94 | 2.82 | - | 7.98 |
| Ethanol | 30 | 20 | 4.34 | ± | 2.36 | 3.97 | 1.51 | - | 9.85 |
| Acetaldehyde | 30 | 20 | 5.24 | ± | 4.17 | 3.53 | 1.56 | - | 15.15 |
| Butanone | 30 | 20 | 0.98 | ± | 1.05 | 0.71 | 0.00 | - | 3.63 |
| Isoprene | 3 | 5 | 0.11 | ± | 0.07 | 0.09 | 0.06 | - | 0.24 |
| α-Pinene | 3 | 5 | 0.11 | ± | 0.12 | 0.05 | 0.03 | - | 0.30 |
| β-Pinene | 3 | 5 | 0.09 | ± | 0.07 | 0.06 | 0.02 | - | 0.18 |
| Σ measured NMVOC | NA | NA | 48 | ± | 19 | 45 | 25 | - | 83 |





**Table 3:** Comparison of mean concentrations of select volatile compounds measured during this study with those measured during prior studies in the Kathmandu Valley and cities in South and Southeast Asia.  Units are ppbv.

| Location and dates<br>Compound | Kathmandu, Nepal<br>April 2015 | Kathmandu, Nepal<br>Dec. 2012 - Jan. 2013 | Kathmandu, Nepal<br>April 2013 | Mohali, India<br>May 2012 | Karachi, Pakistan<br>Dec. 1998 - Jan. 1999 | Lahore, Pakistan<br>Dec. 2012 | Singapore<br>Aug.-Nov. 2012 |
|---|---|---|---|---|---|---|---|
| CO | 770 (750) | 832 (422)[a,b] | 700 (-[c]) | 567 (293) | 1600 (1300) | 4860 (690)[d] | 280 (11) |
| $CH_4$ | 2000 (80) | 2550 (120)[a] | 2183 (252) | - | 6300 (4700) | 5380 (440)[d] | 1822 (6) |
| Acetaldehyde | 5.2 (4.2) | 8.8 (4.6) | - | 6.7 (3.7) | - | - | - |
| Methyl chloride | 0.9 (0.2) | - | - | - | 2.7 (1.5) | - | - |
| Methanol | 4.4 (1.7) | 7.4 (1.3) | - | 37.5 (17.9) | - | - | - |
| Ethanol | 4.3(2.4) | 1.6 (0.8) | - | - | - | - | - |
| Propene | 1.9 (1.0) | 4.0 (1.2) | - | - | 5.5 (5.3) | 18.3 (3.0)[d] | 0.8 (0.2) |
| Benzene | 0.9 (0.5) | 2.7 (1.2) | - | 1.7 (1.5) | 5.2 (4.5) | 28.2 (4.8)[d] | 0.58 (0.06) |
| Toluene | 1.1 (0.5) | 1.5 (0.4) | - | 2.7 (2.9) | 7.1 (7.6) | 32.4 (6.0)[d] | 1.8 (0.3) |
| Xylenes | 1.6 (0.7) | 1.0 (0.3) | - | 2.0 (2.2) | 4.2 (2.4) | 23.2 (3.5)[d] | 0.9 (0.1) |
| Isoprene | 0.11 (0.07) | 1.1 (0.2) | - | 1.9 (0.9) | 0.8 (1.1) | 1.3 (0.2)[d] | 0.27 (0.02) |
| Number of samples | 9 | NA[e] | NA[f] | NA[e] | 78 | 41 | 85 |
| Reference | This study | (Sarkar et al., 2016) | (Mahata et al., 2018) | (Sinha et al., 2014) | (Barletta et al., 2002) | (Barletta et al., 2017) | (Barletta et al., 2017) |

a) Data from Bhardwaj et al. (2018); b) monthly average for January 2013; c) dash denotes data not available; d) standard error, e) NMVOC were measured continuously by PTR-MS, f) CO and $CH_4$ were measured continuously





**Table 4:** Mean (± standard deviation) concentrations of reactive gases, particulate phase inorganic ions, and percent of these species in gas phase. Total concentrations correspond to the sum of gas-phase and $PM_{10}$.

| | Gas-phase (nmol m⁻³) | | PM₂.₅ (nmol m⁻³) | | PM₁₀ (nmol m⁻³) | | Total (nmol m⁻³) | | % in gas phase | | |
| --- | --- | --- | --- | --- | --- | --- | --- | --- | --- | --- | --- |
| | Day | Night | Day | Night | Day | Night | Day | Night | Day | Night | Total |
| $NH_3$ or $NH_4^+$ | 961 (173) | 969 (228) | 272 (40) | 533 (223) | 290 (55) | 610 (251) | 1200 (155) | 1540 (410) | 76 (5) | 61 (7) | 68 (10) |
| $HNO_3$ or $NO_3^-$ | 23 (14) | 1.7 (0.9) | 34 (22) | 64 (38) | 89 (60) | 122 (60) | 114 (65) | 124 (60) | 23 (12) | 2 (1) | 12 (14) |
| $HCl$ or $Cl^-$ | 47 (43) | 24 (24) | 7.0 (8.0) | 91 (63) | 17 (14) | 113 (67) | 43 (31) | 130 (76) | 65 (33) | 15 (17) | 35 (34) |
| $SO_2$ or $SO_4^{2-}$ | 354 (198) | 1023 (344) | 97 (16) | 159 (55) | 115 (19) | 181 (57) | 405 (164) | 1180 (387) | 67 (14) | 84 (5) | 76 (13) |
| $Br^a$ | 0.4 (0.1) | 0.7 (0.3) | - | - | - | - | - | - | NA | NA | NA |

a) Bromine concentrations below detection limit were replaced with the limit of detection (LOD)/√2 in calculating average and standard deviation.






**Table 5:** PM$_{2.5}$ mass concentrations, OC, EC, inorganic ions, and organic species measured at Bode in the Kathmandu Valley.

| Species | Overall | | Daytime (8:00 am - 5:30 pm) | | Nighttime (6:00 pm – 7:30 am) | |
|---|---|---|---|---|---|---|
| | Mean ± std. dev. | Median | Mean ± std. dev. | Range | Mean ± std. dev. | Range |
| **PM$_{2.5}$ mass** (µg m$^{-3}$) | 68.2 ± 34.7 | 65.8 | 54.2 ± 18.0 | 30.0-94.7 | 83.3 ± 42.3 | 32.9-207 |
| **OC** (µg C m$^{-3}$) | 17.6 ± 9.6 | 15.7 | 14.2 ± 5.8 | 7.9-30.7 | 20.8 ± 11.4 | 8.7-57.3 |
| **EC** (µg m$^{-3}$) | 9.0 ± 6.4 | 7.3 | 5.6 ± 3.4 | 2.3-12.8 | 12.2 ± 7.0 | 4.4-30.8 |
| **Inorganic ions** (µg m$^{-3}$) | | | | | | |
| Ammonium | 5.8 ± 2.8 | 5.7 | 3.9 ± 0.9 | 2.4-5.8 | 7.6 ± 2.9 | 3.0-15.6 |
| Sodium | 0.10 ± 0.09 | 0.08 | 0.10 ± 0.10 | <0.02-0.36 | 0.10 ± 0.07 | <0.02-0.23 |
| Potassium | 0.63 ± 0.30 | 0.59 | 0.53 ± 0.17 | 0.28-0.80 | 0.72 ± 0.36 | 0.22-1.8 |
| Calcium | 0.65 ± 0.50 | 0.51 | 0.73 ± 0.63 | 0.03-1.64 | 0.56 ± 0.35 | 0.10-1.07 |
| Magnesium | 0.04 ± 0.03 | 0.04 | 0.05 ± 0.04 | <0.02-0.11 | 0.03 ±0.02 | <0.02-0.05 |
| Nitrate | 2.7 ± 1.7 | 2.5 | 2.3 ± 1.7 | 0.1-6.2 | 3.1 ± 1.7 | 0.4-8.0 |
| Sulfate | 10.2 ± 3.6 | 9.7 | 8.1 ± 2.0 | 5.6-13.0 | 12.2 ± 3.8 | 5.6-22.1 |
| Chloride | 1.52 ± 1.67 | 0.76 | 0.281 ± 0.281 | 0.003-0.757 | 2.67 ± 1.60 | 0.641-6.70 |
| Fluoride | 0.05 ± 0.02 | 0.05 | 0.04 ± 0.02 | <0.02-0.08 | 0.04 ± 0.02 | <0.02-0.10 |
| **Organic species** (ng m$^{-3}$) | | | | | | |
| Levoglucosan | 1230 ± 1154 | 912 | 843 ± 424 | 328-1671 | 1588 ± 1486 | 425-5910 |
| Cholesterol | 2.5 ± 6.1 | 1.1 | 1.0 ± 1.4 | <0.1-4.6 | 3.9 ± 8.3 | <0.1-32.3 |
| Stigmasterol | 3.4 ± 3.1 | 2.3 | 3.0 ± 3.6 | <0.1-13.0 | 3.7 ± 2.7 | 0.9-8.8 |
| $\beta$-Sitosterol | 9.4 ± 10.1 | 7.0 | 6.1 ± 7.2 | 0.03-26.2 | 12.6 ± 11.6 | 3.5-46.8 |
| Campesterol | 2.8 ± 4.5 | 1.8 | 1.1 ± 1.9 | 0.2-6.5 | 4.5 ± 5.5 | 0.4-20.0 |
| Coprostanol | 0.6 ± 1.3 | 0.40 | 0.4 ± 0.4 | <0.4-1.6 | 0.9 ± 1.8 | <0.4-6.7 |
| Stigmastanol | 1.0 ± 0.8 | 0.90 | 0.9 ± 0.8 | <0.4-2.5 | 1.2 ± 0.7 | <0.4-2.7 |
| **PAHs** | | | | | | |
| Phenanthrene | 1.1 ± 1.3 | 0.70 | 0.56 ± 0.38 | 0.11-1.35 | 1.70 ± 1.63 | 0.44-6.37 |
| Anthracene | 0.32 ± 0.36 | 0.17 | 0.20 ± 0.18 | 0.04-0.62 | 0.43 ± 0.45 | 0.12-1.64 |
| Fluoranthene | 3.9 ±4.4 | 2.5 | 1.72 ± 1.05 | 0.47-3.95 | 5.95 ± 5.26 | 1.61-20.4 |
| Pyrene | 4.2 ± 4.8 | 2.6 | 1.84 ± 1.13 | 0.46-4.30 | 6.39 ± 5.86 | 1.78-22.8 |
| Methylfluoranthene | 1.1 ± 1.2 | 0.91 | 0.57 ± 0.35 | 0.11-1.30 | 1.66 ± 1.43 | 0.41-5.15 |
| Benzo(ghi)fluoranthene | 4.3 ± 4.9 | 3.1 | 1.74 ± 1.03 | 0.53-4.08 | 6.73 ± 5.86 | 2.27-21.2 |
| Cyclopenta(cd)pyrene | 1.7 ± 1.9 | 1.3 | 0.85 ± 0.50 | 0.28-1.70 | 2.55 ± 2.27 | 0.86-8.40 |
| Benz(a)anthracene | 3.2 ± 3.9 | 2.0 | 1.15 ± 0.70 | 0.35-2.75 | 5.10 ± 4.65 | 1.42-16.0 |





| | | | | | |
|---|---|---|---|---|---|
| Chrysene | 4.9 ± 5.4 | 3.0 | 1.98 ± 1.04 | 0.78-4.55 | 7.56 ± 6.42 | 2.28-23.0 |
| 1-Methylchrysene | 0.64 ± 0.76 | 0.41 | 0.21 ± 0.17 | <0.03-0.59 | 1.05 ± 0.87 | 0.26-3.35 |
| Retene | 0.28 ± 0.83 | 0.06 | 0.08 ± 0.02 | <0.1-0.15 | 0.47 ± 1.14 | <0.1-4.38 |
| Benzo(b)fluoranthene | 5.8 ± 6.2 | 4.3 | 2.69 ± 1.67 | 0.67-6.59 | 8.60 ± 7.53 | 2.46-27.1 |
| Benzo(k)fluoranthene | 6.0 ±6.2 | 4.1 | 2.37 ± 1.59 | 0.55-6.52 | 9.35 ± 7.10 | 3.43-26.5 |
| Benzo(j)fluoranthene | 1.2 ± 1.2 | 1.0 | 0.50 ± 0.34 | 0.10-1.23 | 1.85 ± 1.36 | 0.65-5.79 |
| Benzo(e)pyrene | 4.5 ± 4.4 | 3.4 | 2.02 ± 1.10 | 0.73-4.10 | 6.79 ± 5.10 | 2.20-19.9 |
| Benzo(a)pyrene | 4.7 ± 5.4 | 3.1 | 1.98 ± 1.19 | 0.64-4.71 | 7.20 ± 6.59 | 2.23-24.7 |
| Perylene | 0.9 ± 0.9 | 0.76 | 0.46 ± 0.27 | 0.19-1.08 | 1.28 ± 1.08 | 0.43-4.19 |
| Indeno(1,2,3-cd)pyrene | 8.9 ± 8.9 | 6.5 | 3.94 ± 2.71 | 1.10-10.27 | 13.54 ± 10.17 | 4.67-41.4 |
| Benzo(ghi)perylene | 6.6 ± 6.3 | 5.0 | 3.32 ± 1.69 | 1.26-6.96 | 9.59 ± 7.54 | 3.47-28.7 |
| Dibenz(ah)anthracene | 1.4 ± 1.4 | 1.2 | 0.76 ± 0.52 | 0.16-1.81 | 1.96 ± 1.73 | 0.66-6.85 |
| Picene | 3.5 ± 3.9 | 2.6 | 1.41 ± 0.92 | 0.34-3.12 | 5.47 ± 4.52 | 1.78-18.5 |
| 1,3,5-Triphenylbenzene | 0.79 ± 0.63 | 0.62 | 0.68 ± 0.70 | 0.25-2.94 | 0.89 ± 0.56 | 0.28-2.31 |
| **∑ PAHs** | **69.9 ± 73.0** | **43.7** | **31.0 ± 17.8** | **10.3-70.6** | **106 ± 87** | **36.2-313** |
| **Hopanes** | | | | | | |
| $17\alpha(H)$-22,29,30-trisnorhopane | 0.22 ± 0.23 | 0.17 | 0.15 ± 0.14 | <0.02-0.56 | 0.29 ± 0.26 | 0.06-1.13 |
| $17\beta(H)$-$21\alpha(H)$-30-norhopane | 0.40 ± 0.38 | 0.28 | 0.19 ± 0.13 | <0.02-0.55 | 0.59 ± 0.43 | 0.09-1.90 |
| $17\alpha(H)$-$21\beta(H)$-hopane | 0.34 ± 0.38 | 0.25 | 0.26 ± 0.20 | 0.02-0.63 | 0.43 ± 0.49 | 0.06-2.00 |
| **∑Hopanes** | **0.95 ± 0.93** | **0.72** | **0.56 ± 0.32** | **0.02-1.36** | **1.31 ± 1.17** | **0.24-5.03** |
| **SOA tracers** | | | | | | |
| *cis*-Pinonic acid | 4.5 ± 1.9 | 4.4 | 6.1 ± 1.3 | 4.4-7.9 | 3.0 ± 0.8 | 2.0-4.6 |
| $\beta$-Caryophyllinic acid | 18.7 ± 9.4 | 15.7 | 17.3 ± 8.0 | 10.0- 40.0 | 19.9 ± 12.1 | 6.7-51.7 |
| Phthalic acid | 60.8 ± 22.2 | 60.1 | 69.6 ± 20.7 | 45.5- 113 | 52.7 ± 21.1 | 28.2-91.9 |
| 4-Methylphthalic acid | 9.3 ± 4.2 | 8.9 | 11.7 ± 3.7 | 8.1- 19.2 | 7.1 ± 3.3 | 3.5-13.7 |

std. dev. = standard deviation



**Table 6**: Relative primary and secondary source contributions to $PM_{2.5}$ OC during day and night periods in the Kathmandu Valley. Source contributions during day and night periods were compared and p-values < 0.05 indicate significant differences at the 95% confidence interval.

| Sources | Overall (%) | Day (%) | Night (%) | p-value |
|---|---|---|---|---|
| Garbage burning | 18.1 ± 4.5 | 19.0 ± 4.2 | 17.2 ± 4.8 | 0.34 |
| Open biomass burning | 17.2 ± 9.5 | 17.3 ± 9.6 | 17.1 ± 9.8 | 0.96 |
| Gasoline & diesel engines | 18.0 ± 9.2 | 14.0 ± 7.3 | 21.8 ± 9.5 | 0.03* |
| Coal combustion | 5.0 ± 2.3 | 4.0 ± 2.0 | 5.9 ± 2.3 | 0.04* |
| Vegetative detritus | 1.6 ± 0.9 | 1.8 ± 1.1 | 1.3 ± 0.6 | 0.12 |
| α-Pinene SOC | 0.13 ± 0.07 | 0.19 ± 0.05 | 0.07 ± 0.03 | <0.001* |
| β-Caryophyllene SOC | 4.6 ± 1.5 | 5.3 ± 1.6 | 4.0 ± 1.1 | 0.05 |
| Naphthalene SOC | 9.8 ± 4.0 | 13.0 ± 3.0 | 7.0 ± 2.5 | <0.001* |
| Methylnaphthalene SOC | 0.25 ± 0.12 | 0.36 ± 0.08 | 0.15 ± 0.05 | <0.001* |
| Other OC | 25.4 ± 16.6 | 25.2 ± 13.9 | 25.6 ± 18.0 | 0.94 |



**Figure 1**: Non-water mass concentrations of (a) $PM_{2.5}$ and (b) $PM_{10}$ during daytime and nighttime periods and mass contributions from OC, EC, and inorganic ions. OC and EC were not measured in $PM_{10}$ samples and were assumed to be the same as $PM_{2.5}$ for mass balance purposes. The remaining mass of $PM_{2.5}$ includes elements associated with OC (hydrogen, oxygen, nitrogen, etc.), metals, and other unmeasured species. Error bars represent propagated analytical
uncertainties. The $PM_{2.5}$ mass was not quantified on 12 April nor 13 April as described in section 2.2.

**Figure 2:** Diurnal trends in average total $PM_1$ mass and concentrations of non-refractory inorganic species measured with the AMS, average BC measured with the aethalometer, and
average wind speed at Bode on 13 and 16 to 24 April 2015. The shaded region represents the duration for nighttime filter collection and the unshaded region represents the duration for daytime filter collection.

**Figure 3:** (a) Temperature and RH; (b) liquid water content (LWC) of $PM_{2.5}$ and $PM_{10}$; (c) gas-
phase $NH_3$ and particulate $NH_4^+$ associated with $PM_{2.5}$ and $PM_{10}$; (d) gas-phase $HNO_3$ and particulate $NO_3^-$ associated with $PM_{2.5}$ and $PM_{10}$; (e) gas-phase $HCl_{calc}$ and particulate $Cl^-$ associated with $PM_{2.5}$ and $PM_{10}$; (f) gas-phase $SO_2$ and particulate $SO_4^{2-}$ associated with $PM_{2.5}$ and $PM_{10}$; (g) detectable concentrations of volatile inorganic Br; (h) equilibrium aerosol solution pH calculated from the measured gas-aerosol phase partitioning of $NH_3$ with $PM_{2.5}$ and $PM_{10}$ and
$HNO_3$ with $PM_{2.5}$ and $PM_{10}$. Vertical gridlines denote local midnight.

**Figure 4**: Daytime and nighttime concentrations of: (a) 1,3,5-triphenylbenzene, (b) levoglucosan, (c) picene, (d) $17\alpha(H)$-22,29,30-trisnorhopane in red, $17\beta(H)$-$21\alpha(H)$-30-norhopane in light green, and $17\alpha(H)$-$21\beta(H)$-hopane in yellow, (e) stigmastanol, (f) cholesterol,
and (g) *cis*-pinonic acid in $PM_{2.5}$ in the Kathmandu Valley, Nepal. Error bars represent analytical uncertainties propagated from measurements. Measurements below the instrumental detection limits are marked as ND.

**Figure 5**: Apportionment of primary and secondary sources for $PM_{2.5}$ OC based on CMB
modeling. Missing primary sources on 13-N and 18-D of April are marked with stars (see section 3.5).

**Figure 6**: Sensitivity of CMB model results to the input source profiles: (a) sensitivity of garbage burning contributions to $PM_{2.5}$ OC to the garbage burning profile and (b) sensitivity of biomass burning contributions to $PM_{2.5}$ OC to biomass burning profiles.




Figure 1

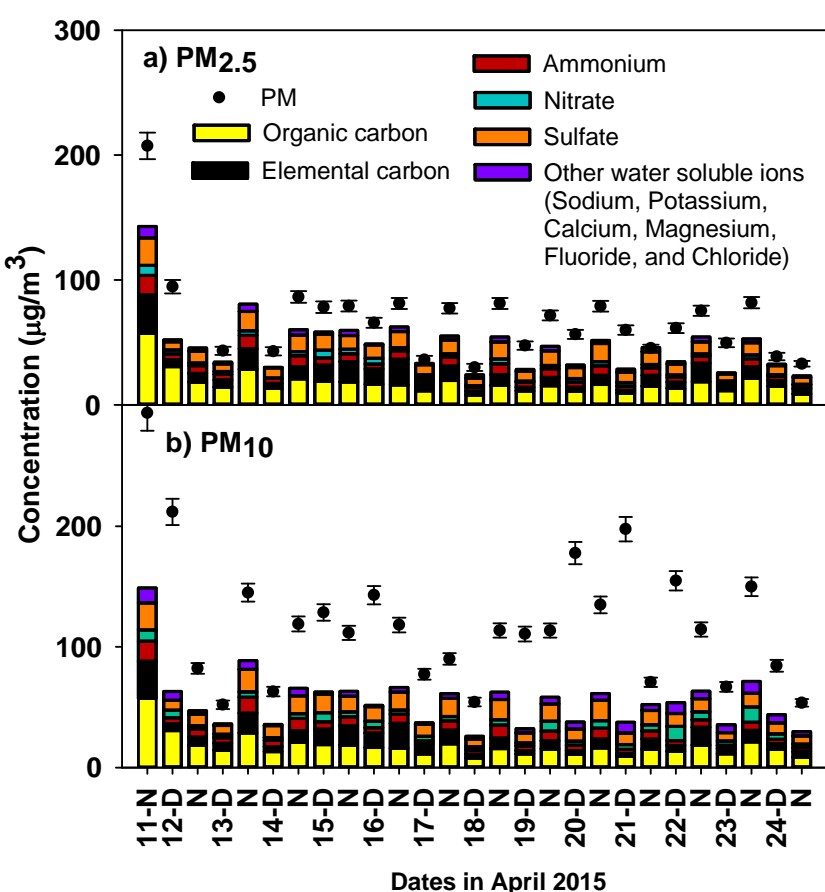



Figure 2

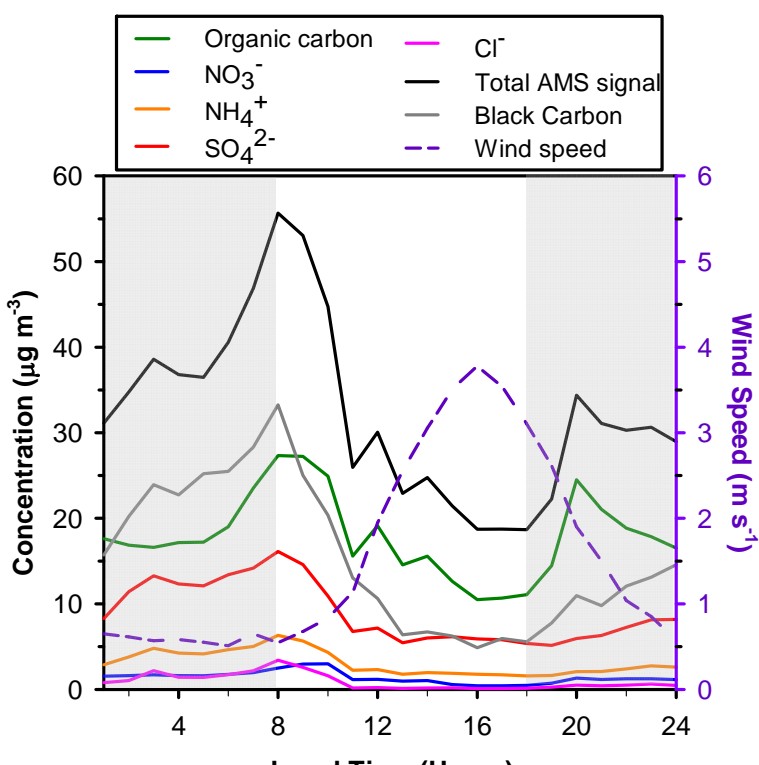





Figure 3

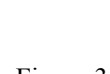

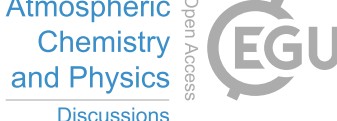

Figure 4


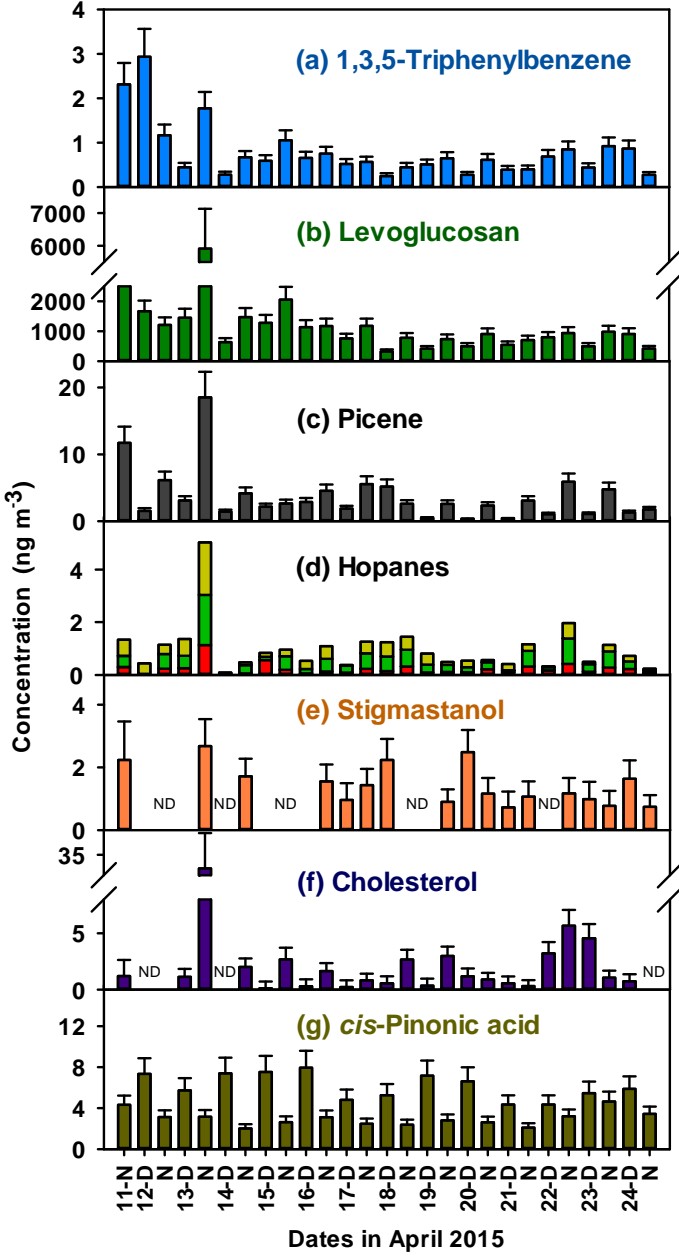





Figure 5

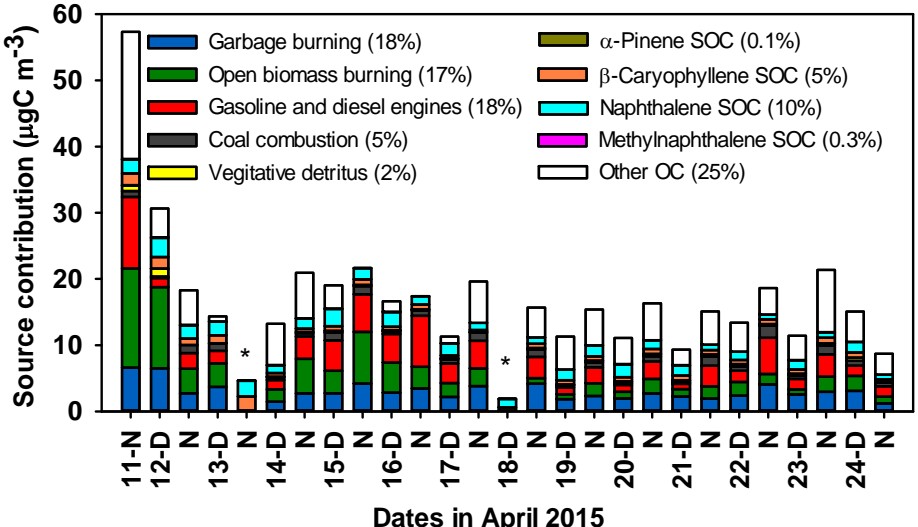



1380        Figure 6

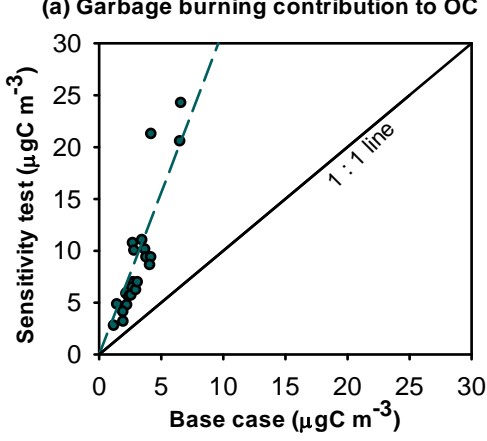

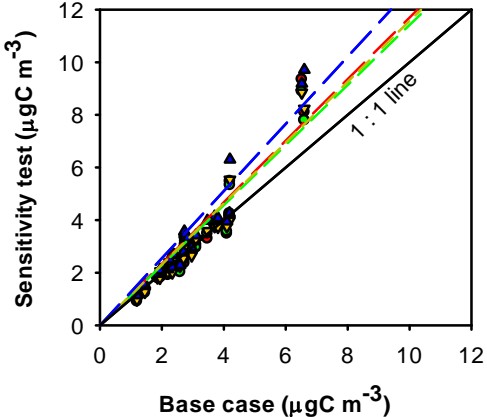