# Peer review of "Ambient air quality in the Kathmandu Valley, Nepal during the pre-monsoon"

_Atmospheric Chemistry and Physics, 2019_

## Referee Comment (RC1) · Anonymous Referee #1 · 24 Jun 2019

This manuscript by Md. Robiul et al. comprehensively analyzed the concentrations and sources of non-methane volatile organic compounds, organic tracers and carbonacousous aerosols at Bode, Nepal. The results highlight that primary sources, including garbage and biomass burning, vehicle emissions, are the dominant sources of air pollutants at Bode. Meanwhile, the diurnal variation of meteorological condition (e.g., atmospheric boundary layer) may also accounts partially for the diel trend of particle abundance. Results in this study could provide insights into the chemical composition and source characteristic of air pollutant at Nepal and benefit the related modeling work. I recommend for publication of this manuscript after a minor revision.

[Figure]

Specific comments 1. Line 180-186, the authors mentioned the general PM1 observations by AMS will be currently used to provide higher time resolution context for the filter measurements discussed in detail. However, I didn't see the detail in the text. 2. Line 383, you listed "CO2, CO, CH4, and NMVOCs" in the title, but I didn't see the discussion of CO2 and CO. 3. Line 332-339, the description of organic species analysis in PM2.5 by GC-MS was too simple, please add the details, i.e., internal recovery standards, authentic standards, organic reagents, the GC temperature program, reproducibility, method detection limits. 4. Line 670-672, I am not quite understanding what the authors mean that "dung burning is not common in the Kathmandu Valley and its outskirts, dung is a more widely used fuel in rural areas of southern Nepal and India and may contribute to the observed dung burning tracers", you means the dung burning tracer was from long-range transport? 5. Line 420, Line 680-685, the authors speculated the lower isoprene was caused by the unusually cold weather during spring 2015, however, the temperature was 12 to 28 °C according to Fig. 3 (a), which was higher than winter. The methyltetrols are mainly formed under low-NOx or NOx -free conditions. Therefore, could you please give some more reasonable explanation? 6. Line 694-695, do you think there is no transport of air masses passing over Kathmandu during the nighttime? 7. Section 3.3, The main pollution events during the 9-day festival affected the results of the CMB source apportionment, what's the source contribution excluding the main pollution events? 8. Section 3.4, it's better to add the discussion of reactive trace gases during the pollution events, which will make the MS full of logicality and tightness. 9. Line 815, why there is vegetative detritus contributing to EC? As we all know, the vegetative detritus is contributor to OC. Could you provide some other explanation or add some references? 10. Line 860-884, the conclusions were a little simple, the authors did long discussion of the data about the ambient air quality in the Kathmandu Valley from the concentrations and sources of particulate matter and trace gases, however, you only pointed out the garbage burning, biomass burning, and vehicle emissions are potential targets for emissions reductions to reduce ambient PM2.5 in Kathmandu Valley. I welcome seeing a more informative summary in the conclusions

section.

---

## Referee Comment (RC2) · Anonymous Referee #2 · 12 Oct 2019

General Comments

The paper by Islam et al. describes a slightly more than two week set of trace gas and aerosol measurements in the Kathmandu Valley in spring (April) of 2005. The measurements would have extended to a longer period of time if not interrupted by an earthquake that year. Filter based measurements were available at twice daily resolution for a set of speciated compounds, while higher time resolution particle data were available from an aerosol mass spectrometer. The analysis attributes sources of organics that contributed the major fraction of the particle phase, together with some analysis of inorganic aerosol thermodynamic properties and partitioning of major inorganic ions

between gas and condensed phases.

The paper is a useful contribution to the literature in a relatively polluted but under-sampled region. Publication is recommended following attention to the comments below.

Specific comments

Line 92: "A satellite-derived . . ."

Lines 94-96: This is not a closed budget or even a "major" components budget since most of the mass is not assigned.

Line 126: A figure with a map indicating the location of the site within the Kathmandu Valley, as well as the location of this valley within a wider geographic region would help with context.

Line 161 (and elswehre): $NO_3$ and $Cl$ should be indicated as anions

Lines 259-274: The calculation appears to consider only the inorganic component of the particles. Actual liquid water content and deliquescence state should depend on the organic component as well. Have organics been excluded from the calculation? Please comment here or in the section at the end on limitations of the calculations.

Lines 286-295: Description of how HCl can be estimated from measurement of total chloride (presumably meaning gas + aerosol phase) that were at or below the detection limits is not clear. Please clarify how in the absence of a total chloride measurement but a model prediction of the aerosol pH it is possible to estimate gas phase HCl. This is likely just a wording / clarity issue as the data in Figure 3 are not consistent with a measurement that is below detection limit.

Line 320: This appears to be the first reference to a figure from the text. Normally they are called out in order rather than beginning with figure 3.

Line 436 and Table 3: The meaning of "reasonable" is not clear here. Is this intended
to indicate that the measurements are accurate ? If so, the analytical descriptions above are sufficient. More to the point, however, the comparisons of absolute mixing ratios from one place to another in Table 3 are only partially informative due to the reasons cited in the preceding paragraph that they are drawn from different seasons with likely very different meteorology and boundary layer depths. A better comparison would involve ratios of selected compounds to tracers such as CO to give a sense for the contributions to emissions in different locations.

Section 3.1.2: Here specific ratios are discussed, which is helpful, but not given in table format but rather only as in-text description? Suggest combining this information with that in table 3 to make the section more readable.

Line 574-576: Given the variability in other inorganic components are mainly ascribed to meteorology and transport rather than chemistry, the simplest explanation for the apparent diel cycle in Br species would also be meteorology.

Lines 582-599: While consistent with the recent literature and the observation of large amounts of soluble halides, the statements here are qualitative only but could be made more quantitative. For example, with data for O3, NO2 and related species, rates of the relevant gas and heterogenous phase processes could be estimated. If the supporting data for such a calculation does not exist, the paragraph should state this as a justification for not doing the calculation.

Line 747-749: Coal combustion is analyzed as a source of PM2.5, especially at night. Similarly, this could be mentioned as an explanation for the inorganic ions, particularly the halides, in the preceding section.

Line 770-774: A diel dependence of ozone would also support the analysis of photo-chemical vs nighttime contributions to secondary organics if the data are available.

---

## Author Comment (AC1) · 22 Nov 2019

**Response to Interactive Comments on Atmospheric Chemistry and Physics Discussions (ACPD)**

https://doi.org/10.5194/acp-2019-333-RC2, 2019

**Ambient air quality in the Kathmandu Valley, Nepal during the pre-monsoon: Concentrations and sources of particulate matter and trace gases**

Md. Robiul Islam[1], Thilina Jayarathne[1], Isobel J. Simpson[2], Benjamin Werden[3], John Maben[4],

Ashley Gilbert[1], Puppala S. Praveen[5], Sagar Adhikari[5,6], Arnico K. Panday[5], Maheswar Rupakheti[7],

Donald R. Blake[2], Robert J. Yokelson[8], Peter F. DeCarlo[3,9], William C. Keene[4], Elizabeth A.

Stone[1,10]

[1]University of Iowa, Department of Chemistry, Iowa City, IA, USA
[2]University of California-Irvine, Department of Chemistry, Irvine, CA, USA
[3]Drexel University, Department of Civil, Architectural, and Environmental Engineering, Philadelphia, PA, USA
[4]University of Virginia, Department of Environmental Sciences, Charlottesville, VA, USA
[5]International Centre for Integrated Mountain Development (ICIMOD), Lalitpur, Nepal
[6]MinErgy Pvt. Ltd, Lalitpur, Nepal
[7]Institute for Advanced Sustainability Studies, Potsdam, Germany
[8]Department of Chemistry, University of Montana, Missoula, MT, USA
[9]Drexel University, Department of Chemistry, Philadelphia, PA, USA
[10]Department of Chemical and Biochemical Engineering, University of Iowa, Iowa City, IA, USA

Received: 05 Apr 2019 – Accepted for review: 12 Apr 2019 – Discussion started: 08 May 2019

Correspondence to: E. A. Stone (betsy-stone@uiowa.edu)

Published by Copernicus Publications on behalf of the European Geosciences Union.

*Anonymous referee #1 general comments:* *"This manuscript by Md. Robiul et al. comprehensively analyzed the concentrations and sources of non-methane volatile organic compounds, organic tracers and carbonaceous aerosols at Bode, Nepal. The results highlight that primary sources, including garbage and biomass burning, vehicle emissions, are the dominant sources of air pollutants at Bode. Meanwhile, the diurnal variation of meteorological condition (e.g., atmospheric boundary layer) may also accounts partially for the diel trend of particle abundance. Results in this study could provide insights into the chemical composition and source characteristic of air pollutant at Nepal and benefit the related modeling work. I recommend for publication of this manuscript after a minor revision."*

**Response to referee #1 general comments:** We thank the referee for the careful review of the manuscript. Specific comments are addressed point-by-point below.

*Anonymous referee #1 specific comment 1:* *"Line 180-186, the authors mentioned the general PM1 observations by AMS will be currently used to provide higher time resolution context for the filter measurements discussed in detail. However, I didn't see the detail in the text."*

**Response to referee #1 specific comment 1:** To clarify to the reader the location of our discussion of the higher-time resolution AMS measurements and filter-based measurements of PM, we have added the following parenthetic information to the end of this sentence at line 186: "…(see section 3.2.1 for a discussion of PM mass and 3.2.3 for sulfate concentrations)."

*Anonymous referee #1 specific comment 2:* *"Line 383, you listed "CO2, CO, CH4, and NMVOCs" in the title, but I didn't see the discussion of CO2 and CO."*

**Response to referee #1 specific comment 2:** As suggested by the referee, we have expanded our discussion to include $CO_2$ and CO at the end of section 3.1.1. Accordingly, we have revised the subsection title to "***Abundance of VOC, CO_2, and CO***"

"$CO_2$ concentrations in Kathmandu are elevated above the global background (Table 2). CO (like the NMVOCs and PM) is an excellent indicator of air pollution levels and is derived from combustion rather than solvents or secondary sources. As shown in Table 3, the pattern of CO enhancements relative to other studies is similar to the pattern for most NMVOCs. The three studies in Kathmandu during the dry season have similar values to each other. The importance of combustion as a source of air pollutants, particularly PM, is consistent with the carbon mass balance source apportionment results discussed in section 3.3."

*Anonymous referee #1 specific comment 3:* *"Line 332-339, the description of organic species analysis in PM2.5 by GC-MS was too simple, please add the details, i.e., internal recovery standards, authentic standards, organic reagents, the GC temperature program, reproducibility, method detection limits."*

**Response to referee #1 specific comment 3:** We are happy to provide the details of the organic species analysis. As the current manuscript is very long, we choose to provide the additional details in the supporting information of the manuscript (S-1).

At line 342, the following text has been added: "Details of the extraction process and GC temperature program are provided in the supplemental information (S-1).

The information added to the supporting information is:

**"SI section S-1: Detailed description of the extraction and analysis of organic species.** Prior to extraction, filters were spiked with isotopically labelled internal standards: pyrene-$D_{10}$, benz(a)anthracene-$D_{12}$, cholestane-$D_4$, pentadecane-$D_{32}$, eicosane-$D_{42}$, tetracosane-$D_{50}$, triacontane-$D_{62}$, dotriacontane-$D_{66}$, hexatriacontane-$D_{74}$, levoglucosan-$^{13}C_6$ and cholesterol $D_6$. Filters were extracted using two 20 mL portions of hexanes (Optima, 99.9%) then two 20 mL portions of acetone (CHROMASOLV®, for HPLC, ≥99.9%) by ultra-sonication (Branson 5510) at 42 kHz frequency and 20-25 ºC temperature for 10 min each as described in Al-Naiema et al. (2015). After extraction, the extract was combined and the volume was reduced to ~4 mL under high purity $N_2$ (PRAXAIR Inc.; Zymark Turbo-Vap II, LV, Caliper Life Science). The extract was filtered using a 0.2 µm PTFE filter (Whatman, GE Health Care Life Sciences) and further evaporated to ~1 mL under high purity $N_2$ (Reacti-Vap I, Thermo Scientific). Extracts were stored in amber vials at -20 ºC. Immediately prior to analysis, extracts were evaporated to a final volume of 100 µL under high purity $N_2$.

Samples were directly analyzed for nonpolar organic species, including polycyclic aromatic hydrocarbons, hopanes, and alkanes. Hydroxyl and carboxylic acid-bearing analytes, including levoglucosan, methyltetrols, and phthalic acids underwent silylation derivatization prior to analysis as described by Stone et al. (2012) to convert active hydrogen atoms to trimethylsilyl (TMS) groups (Nolte et al., 2002). For this derivitization, a 10 µL aliquot of the extract was dried at 30 ºC under gentle nitrogen flow, then 10 µL of pyridine (Burdick & Jackson, Anhydrous) and 20 µL of the silylation agent N,O-bis(trimethylsilyl)trifluoro-acetamide (Fluka Analytical 99%) were added. The mixture was heated at 70 ºC for 3 h before analysis by GC-MS.

Instrumental analysis utilized a gas chromatograph coupled with a mass spectrometer (GC-MS, Agilent Technologies GC-MS 7890A) equipped with an Agilent DB-5 column (30 m x 0.25 mm x 0.25 µm) and electron ionization (EI) source with a temperature program described in Stone et al. (2012). In brief, non-polar organic species were analyzed by injecting 2 µL aliquots to the GC inlet operating in the splitless mode at 300 ºC. The separation was achieved with an initial oven temperature of 65 ºC, held for 10 min, and then ramped at a rate of 10 ºC $min^{-1}$ to 300 ºC and held for 26.5 min. For the analysis of silylated samples, 2.0 mL of each sample was injected to the GC inlet operating in the splitless mode at 270 ºC. The initial GC oven temperature was 84 ºC, held for 1 min, then increased at a rate of 8 ºC $min^{-1}$ to 200 ºC and held for 2 min, and then ramped at a rate of 10 ºC $min^{-1}$ to 300 ºC and held for 15 min. For all analyses, the GC–MS interface was held at 300 ºC, the MS quadrupole and source were operated at 150 ºC and 230 ºC, respectively. Responses of analytes were normalized to the corresponding isotopically-labeled internal standard and quantified with five point linear calibration curves (with correlation coefficients, $R^2$≥0.995)."

We also presented the data regarding reproducibility and method detection limits in Table S1 in the supplemental information, and added the following sentence at line 351. "Reproducibility and method detection limits for all the organic species are presented in Table S1 in the supplemental document."

*Anonymous referee #1 specific comment 4: "Line 670-672, I am not quite understanding what the authors mean that "dung burning is not common in the Kathmandu Valley and its outskirts, dung is a more widely used fuel in rural areas of southern Nepal and India and may contribute to the observed dung burning tracers", you means the dung burning tracer was from long-range transport?"*

**Response to referee #1 specific comment 4:** To clarify that dung burning PM in the Kathmandu Valley was likely derived from long-range transport rather than from local sources, we revised the sentence at line 670-672 to read: "Although dung burning is not common in the Kathmandu Valley and its outskirts, dung is a more widely used fuel in rural areas of southern Nepal and India. Thus, it is expected that most of the dung burning tracers observed at Bode had been transported from other regions."

***Anonymous referee #1 specific comment 5:*** *"Line 420, Line 680-685, the authors speculated the lower isoprene was caused by the unusually cold weather during spring 2015, however, the temperature was 12 to 28 °C according to Fig. 3 (a), which was higher than winter. The methyltetrols are mainly formed under low-NOx or NOx –free conditions. Therefore, could you please give some more reasonable explanation?"*

**Response to referee #1 specific comment 5:** Following the reviewer's suggestion, we have modified sentence at line 189 to include the sampling time: "…. April 2015 before 8:25 or after 18:00 and analyzed…"

We have also added the following sentences at line 422: "Sample collection in early morning and late afternoon also contributed to low isoprene in this study as peak isoprene concentration is typically observed during the midday (Karl et al., 2007). Previous studies report two primary reasons for low isoprene emissions: i) immaturity of leaves, until reaching an age of 23 days (Kuzma and Fall, 1993), and ii) temperatures lower than 35 ºC (Monson et al., 1992)."

The following text has been added to line 685: "The chromatographic data suggested the presence of 2-methylglyceric acid, an isoprene SOA tracer generated under high-NOx conditions; however, this species could not be semi-quantified due to the low recovery (<10%) of structurally-similar hydroxy-acids from the solvent extraction. Nonetheless, these results suggest that in isoprene-derived SOA in the Kathmandu Valley has a larger relative contribution from high-NOx reactions compared to low-NOx. The relative distribution of high- and low-NOx isoprene SOA tracers should be evaluated in future studies."

***Anonymous referee #1 specific comment 6:*** *"Line 694-695, do you think there is no transport of air masses passing over Kathmandu during the nighttime?"*

**Response to referee #1 specific comment 6:** We thank the referee for bringing this point into discussion. The diurnal wind dynamics in the Kathmandu Valley have been previously studied, and described by (Mahata et al., 2017; Panday and Prinn, 2009; Panday et al., 2009; Sarkar et al., 2016) and are briefly described in this manuscript at lines 501-508. The observed wind direction during April 2015 has been added to Figure 2 to better clarify this point. Based on these observations, we do not expect transport of air masses passing over Kathmandu to reach Bode during the nighttime.

[Figure]

*Anonymous referee #1 specific comment 7:* "Section 3.3, The main pollution events during the 9-day festival affected the results of the CMB source apportionment, what's the source contribution excluding the main pollution events?"

**Response to referee #1 specific comment 7:** We thank the referee for bringing this interesting point to our attention. The following text has been added at line 860: "The major sources of PM2.5 OC during 19-24 April after the 9-day festival were still garbage burning (18 ± 4%), biomass burning (11 ± 3%), and gasoline and diesel engines (15 ± 6%). However, contributions from biomass burning (23 ± 10%) and gasoline and diesel burning (21 ± 11%) were relatively higher during the 9-day festival while the contribution from garbage burning (18 ± 5%) remained the same. Meanwhile, the percent contributions from other primary and secondary sources remained very consistent throughout the whole sampling period."

*Anonymous referee #1 specific comment 8:* "Section 3.4, it's better to add the discussion of reactive trace gases during the pollution events, which will make the MS full of logicality and tightness."

**Response to referee #1 specific comment 8:** We agree with the referee and have expanded the discussion in section 3.4 accordingly.

The following sentence has been added at line 839: "Similarly, inorganic gases and other PM$_{2.5}$ species were elevated on 11 and 13 April, by factors of two to four."

The concluding sentence to this paragraph has been revised to include these additional species. Specifically "molecular markers" has been changed to "species."

*Anonymous referee #1 specific comment 9:* "Line 815, why there is vegetative detritus contributing to EC? As we all know, the vegetative detritus is contributor to OC. Could you provide some other explanation or add some references?"

**Response to referee #1 specific comment 9:** We are happy to clarify this point. First, it is important to recognize that vegetative detritus is a negligible source of EC, contributing 0.1% on average. The origin of this EC comes from a small contribution from EC to PM mass in the vegetative detritus source profile adopted from (Hildemann et al., 1991), which has an EC to OC ratio of 0.029. To clarify this information, we added Hildemann et al. (1991) as a reference for vegetative detritus at line 358-363.

*Anonymous referee #1 specific comment 10:* "Line 860-884, the conclusions were a little simple, the authors did long discussion of the data about the ambient air quality in the Kathmandu Valley from the concentrations and sources of particulate matter and trace gases, however, you only pointed out the garbage burning, biomass burning, and vehicle emissions are potential targets for emissions reductions to reduce ambient PM2.5 in Kathmandu Valley. I welcome seeing a more informative summary in the conclusions section."

**Response to referee #1 specific comment 10:** As suggested by the referee we have expanded the conclusion to discuss the significance of other emission sources.

The following sentences have been added at line 873: "The importance of brick kilns to gases and PM in Kathmandu is demonstrated by the elevated concentrations of $SO_2$, $SO_4^{2-}$, $NH_3$, $NH_4^+$, $K^+$ and $Cl^-$ as well as increased coal burning contributions to $PM_{2.5}$ OC at night."

Gasoline evaporation and poorly maintained vehicles as well as some unidentified mixed sources were already mentioned as major contributors of VOCs. Thus, the following sentence has been added at line 877: "Controlling these combustion sources would also reduce emissions of VOCs, $SO_2$, $NO_x$, and reactive halogen species that impact air quality through interrelated gas-phase and multiphase chemical pathways that produce SOA and contribute to aerosol acidity."

The sentence at line 875 is also revised to include "coal combustion".

*Anonymous referee #2 general comments: "The paper by Islam et al. describes a slightly more than two week set of trace gas and aerosol measurements in the Kathmandu Valley in spring (April) of 2015. The measurements would have extended to a longer period of time if not interrupted by an earthquake that year. Filter based measurements were available at twice daily resolution for a set of speciated compounds, while higher time resolution particle data were available from an aerosol mass spectrometer. The analysis attributes sources of organics that contributed the major fraction of the particle phase, together with some analysis of inorganic aerosol thermodynamic properties and partitioning of major inorganic ions between gas and condensed phases. The paper is a useful contribution to the literature in a relatively polluted but undersampled region. Publication is recommended following attention to the comments below."*

**Response to referee #2 general comments:** We thank the referee for the careful review of the manuscript. Specific comments are addressed point-by-point below.

*Anonymous referee #2 specific comment 1: "Line 92: A satellite-derived . . ."*

**Response to referee #2 specific comment 1:** We have corrected this sentence as suggested.

*Anonymous referee #2 specific comment 2: "Lines 94-96: This is not a closed budget or even a "major" components budget since most of the mass is not assigned."*

**Response to referee #2 specific comment 2:** We agree with the referee and have revised this sentence to begin with "Measured components…" rather than "Major components…".

*Anonymous referee #2 specific comment 3: "Line 126: A figure with a map indicating the location of the site within the Kathmandu Valley, as well as the location of this valley within a wider geographic region would help with context."*

**Response to referee #2 specific comment 3:** We thank the referee for this suggestion. A map indicating the location of the site within the Kathmandu Valley, as well as the location of this valley within a wider geographic region is added as figure S1 in the supplemental information.

[Figure]

Figure S1: Location of the Kathmandu Valley in the wider geographic region (a), and the location of Bode, the site of sample collection in the Kathmandu Valley (b).

*Anonymous referee #2 specific comment 4:* *"Line 161 (and elswehre): NO3 and Cl should be indicated as anions."*

**Response to referee #2 specific comment 4:** As indicated on line 160, this paragraph refers explicitly to "soluble reactive trace gases" not ionic $NO_3^-$ and $Cl^-$, which are associated exclusively with deliquesced particles. As reported in the cited papers, multiple chemically distinct inorganic gases that contain $NO_3$, $Cl$, and $Br$ are sampled by alkaline-impregnated filters and thereby contribute to total volatile inorganic $NO_3$, $Cl$, and $Br$, respectively, measured by the technique. The use of "total volatile inorganic" to characterize these data is consistent with terminology employed in the cited papers and reflects the fact that the technique does not speciate these classes of inorganic gases. As noted on lines 161 and 162, available evidence indicates that total volatile inorganic $NO_3$ and $Cl$ are dominated by $HNO_3$ and $HCl$, respectively. Compounds that contribute to total volatile inorganic $Br$ are mentioned explicitly in Section 3.2.3, line 571.

*Anonymous referee #2 specific comment 5:* *"Lines 259-274: The calculation appears to consider only the inorganic component of the particles. Actual liquid water content and deliquescence state should depend on the organic component as well. Have organics been excluded from the calculation? Please comment here or in the section at the end on limitations of the calculations."*

**Response to referee #2 specific comment 5:** We agree with the reviewer. As indicated in the original manuscript and the cited paper (lines 239 to 240), "… E-AIM Model IV … considers particles comprised of $NH_4^+$, $Na^+$, $SO_4^{2-}$, $NO_3^-$, $Cl^-$, and $H_2O$ (Friese and Ebel, 2010) …" In response to the reviewer's recommendation, the following text has been added starting on line 292.

"… As indicated above, E-AIM Model IV evaluates only a subset of major inorganic constituents. Because potential influences of organic matter on aerosol hygroscopic properties are not considered, the modeled estimates of water contents may diverge to some extent from those in ambient air. However, as mentioned above, paired independent estimates of aerosol solution pH based on the phase partitioning of $HNO_3$ and of $NH_3$ and corresponding meteorological conditions measured simultaneously yielded similar results. These two compounds have distinct thermodynamic properties and associated pH-dependent solubilities; the solubility of $HNO_3$ decreases whereas that of $NH_3$ increases with decreasing solution pH. The good agreement between these paired results suggests that estimates of aerosol pH during the campaign were relatively insensitive to potential influences of organic matter on water contents. …"

*Anonymous referee #2 specific comment 6:* *"Lines 286-295: Description of how HCl can be estimated from measurement of total chloride (presumably meaning gas + aerosol phase) that were at or below the detection limits is not clear. Please clarify how in the absence of a total chloride measurement but a model prediction of the aerosol pH it is possible to estimate gas phase HCl. This is likely just a wording / clarity issue as the data in Figure 3 are not consistent with a measurement that is below detection limit."*

**Response to referee #2 specific comment 6:** We did not estimate HCl from total chloride. HCl during each sampling interval (referred to as $HCl_{calc}$ was calculated from the mean $H^+$ activity inferred from the measured phase partitioning of $HNO_3$ and of $NH_3$, the measured $Cl^-$ concentration for $PM_{2.5}$, the thermodynamic properties of HCl, meteorological conditions, and the aerosol LWC, $Cl^-$ activity coefficient, and fraction of ionized $Cl^-$ predicted by E-AIM. The cited paragraph has been clarified as follows:

"Although all concentrations of particulate $Cl^-$ were greater than estimated detection limits, most mixing ratios (75%) for volatile inorganic $Cl$ were less than the detection limit and the balance of measurements

were near the detection limit. Consequently, the phase partitioning of HCl and associated data interpretations were poorly constrained. However, $NH_3$, $HNO_3$, particulate $NH_4^+$, and particulate $NO_3^-$ were present at concentrations well above the corresponding detection limits and, as described in Section 3.2.4 below, the measured gas-aerosol phase partitioning of $NH_3$ and $HNO_3$ yielded paired estimates of aerosol solution pHs that agreed well (generally within ±0.1 to ±0.3 pH units). In the absence of direct reliable measurements of HCl, the equilibrium mixing ratio for HCl during each sampling interval (hereafter referred to as $HCl_{calc}$) was estimated using the thermodynamic approach described above based on the mean $H^+$ activity inferred from the measured phase partitioning of $NH_3$ and of $HNO_3$, the $Cl^-$ concentration for $PM_{2.5}$, the thermodynamic properties of HCl, meteorological conditions, and the aerosol LWC, $Cl^-$ activity coefficient, and fraction of measured particulate $Cl^-$ that was ionized as predicted by E-AIM."

***Anonymous referee #2 specific comment 7:*** *"Line 320: This appears to be the first reference to a figure from the text. Normally they are called out in order rather than beginning with figure 3."*

**Response to referee #2 specific comment 7:** We thank the reviewer for noting this out of order figure number. We removed the text (Fig. 3a) from line 320, as this text is self-explanatory with the data value.

***Anonymous referee #2 specific comment 8:*** *"Line 436 and Table 3: The meaning of "reasonable" is not clear here. Is this intended to indicate that the measurements are accurate? If so, the analytical descriptions above are sufficient. More to the point, however, the comparisons of absolute mixing ratios from one place to another in Table 3 are only partially informative due to the reasons cited in the preceding paragraph that they are drawn from different seasons with likely very different meteorology and boundary layer depths. A better comparison would involve ratios of selected compounds to tracers such as CO to give a sense for the contributions to emissions in different locations."*

**Response to referee #2 specific comment 8:** We have clarified the sentence as suggested by the referee. We have deleted "are reasonable compared to other studies and" from the sentence at line 434.

The purpose of this paragraph is to compare the absolute concentrations of NMVOCs in the Kathmandu Valley with other studies. The concentration of CO is also reported in Table 3, from which ratios of NMVOCs to CO are readily calculated. The ratios of NMVOC are the focus of the discussion in 3.1.2 and additional ratios have been added to Table 3 as suggested in the following comment.

***Anonymous referee #2 specific comment 9:*** *"Section 3.1.2: Here specific ratios are discussed, which is helpful, but not given in table format but rather only as in-text description? Suggest combining this information with that in table 3 to make the section more readable."*

**Response to referee #2 specific comment 9:** As suggested by the reviewer, we have added NMVOC ratios discussed in the text (specifically *i*-pentane/*n*-pentane, ethene/ethyne, and *i*-butane/*n*-butane) to Table 3.

***Anonymous referee #2 specific comment 10:*** *"Line 574-576: Given the variability in other inorganic components are mainly ascribed to meteorology and transport rather than chemistry, the simplest explanation for the apparent diel cycle in Br species would also be meteorology."*

**Response to referee #2 specific comment 10:** We agree with the reviewer and so state in the manuscript that diel variability in transport may have contributed to diel variability in $Br_t$. However, without corresponding measurements of particulate $Br^-$, we think it unreasonable to dismiss potential influences of photochemistry in driving diel variability in $Br_t$. Consequently, we prefer to retain the original text (lines

572-574), which mentions potential influences of both "… a possible diel cycle in multiphase chemical processing of Br and/or systematic variability as a function of transport from different source regions"

***Anonymous referee #2 specific comment 11:*** *"Lines 582-599: While consistent with the recent literature and the observation of large amounts of soluble halides, the statements here are qualitative only but could be made more quantitative. For example, with data for O3, NO2 and related species, rates of the relevant gas and heterogenous phase processes could be estimated. If the supporting data for such a calculation does not exist, the paragraph should state this as a justification for not doing the calculation."*

**Response to referee #2 specific comment 11:** The relevant ancillary data required for a more quantitative assessment were not generated during the campaign. In response to the reviewer's recommendation, we have added the following sentence at line 582: "The lack of relevant ancillary measurements during the period of the campaign precluded a quantitative assessment of the potential impacts of reactive halogens on regional air quality in the Kathmandu Valley." We also edited the following sentence at line 582-586 to make it more readable.

***Anonymous referee #2 specific comment 12:*** *"Line 747-749: Coal combustion is analyzed as a source of PM2.5, especially at night. Similarly, this could be mentioned as an explanation for the inorganic ions, particularly the halides, in the preceding section."*

**Response to referee #2 specific comment 12:** We agree with the reviewer. We have expanded our discussion at line 569 to further emphasize this point. The revised text reads: "…suggesting their co-emission from… brick kilns located within the Kathmandu Valley that impact air masses arriving at Bode at night (Stockwell et al., 2016), and/or garbage burning (Jayarathne et al., 2018)."

***Anonymous referee #2 specific comment 13:*** *"Line 770-774: A diel dependence of ozone would also support the analysis of photochemical vs nighttime contributions to secondary organics if the data are available."*

**Response to referee #2 specific comment 13:** We thank the reviewer for the suggestion. Ozone was measured at Bode during this campaign. Because of the high time resolution measurements and diel variation in ozone concentrations, we believe that it will be most effectively presented with the high-time resolution AMS measurements, which will be the subject of a forthcoming publication. As noted in our response to comment 10 by referee 2, diel variability in regional transport complicates reliable interpretation of corresponding variability in ozone and secondary organics with respect to photochemistry alone.

[revised manuscript text omitted]